# Surface Roughness Uniformity Improvement of Additively Manufactured Channels’ Internal Corners by Liquid Metal-Driven Abrasive Flow Polishing

**DOI:** 10.3390/mi16090987

**Published:** 2025-08-28

**Authors:** Yapeng Ma, Kaixiang Li, Baoqi Feng, Lei Zhang

**Affiliations:** College of Mechanical and Electrical Engineering, Soochow University, Suzhou 215021, China; 20214029004@stu.suda.edu.cn (Y.M.); 20235229013@stu.suda.edu.cn (K.L.); 20234229012@stu.suda.edu.cn (B.F.)

**Keywords:** additive manufacturing, abrasive flow machining (AFM), uniform material removal, liquid metal propulsion

## Abstract

Additive manufacturing (AM) enables the production of complex components but often results in poor surface quality due to its layer-by-layer deposition process. To improve surface finish, postprocessing methods like abrasive flow machining (AFM) are necessary. However, conventional AFM struggles with achieving uniform polishing in intricate regions, especially at internal corners. This study proposes a liquid metal-driven abrasive flow (LM-AF) strategy designed for polishing complex internal channels in AM parts. By combining experimental and numerical simulations, the research investigates surface roughness variations, particularly focusing on the Sa (Arithmetic Average Surface Roughness) parameter. Experimental results show that conventional AFM leaves significant roughness at internal corners compared to adjacent areas. To address this, a hybrid GA-NN-GA (Genetic Algorithm–Neural Network-Genetic Algorithm) optimization model was developed. The model uses a neural network to predict Sa based on key parameters, with genetic algorithms applied for training and optimization. The optimal process parameters identified include a NaOH concentration of 1 mol/L, a voltage of 50 V, abrasive concentration of 10%, and a frequency of 428.3 Hz. With these parameters, LM-AF significantly reduced roughness at internal corners of flow channels, achieving uniformity with Sa values reduced from 25.365 μm to 15.780 μm, from 22.950 μm to 15.718 μm, and from 10.933 μm to 10.055 μm, outperforming traditional AFM methods.

## 1. Introduction

Additive manufacturing (AM) has emerged as a transformative technology for the fabrication of complex components, offering exceptional flexibility in rapid prototyping and virtually no restrictions on part geometry [1,2]. AM not only enables the production of intricate structures [3], but also facilitates the integration of multiple components into a single entity [4,5]. This innovation has significantly reshaped manufacturing strategies for critical parts across various industries, including aerospace [6], automotive [7], and biomedical implants [8]. In particular, AM-fabricated parts with embedded fluidic channels are widely employed in applications such as cooling passages in aircraft engines [9] and conformal cooling systems in injection molds [10]. The internal surface quality of geithese channels plays a critical role in determining the performance and reliability of such systems. However, AM components typically suffer from surface defects caused by staircase effects, powder adhesion, and balling phenomena [11]. For internal fluidic passages, the presence of adhered powders and partially melted particles significantly compromises the as-built surface quality [12], necessitating post-processing to enhance surface performance [13]. Notably, surface roughness inconsistencies-especially at geometrically complex features such as corners-remain a major challenge for conventional abrasive flow machining (AFM). Uneven material removal and surface irregularities in the corner regions have been identified as key limitations that affect the operational efficiency and long-term durability of AM channels [14].

The applications of Abrasive Flow Machining (AFM) have been widely explored to enhance the surface quality and mechanical properties of Additive Manufacturing (AM) parts. Studies on process parameters, such as the number of cycles, abrasive media viscosity, abrasive particle size, and abrasive mass fraction, have demonstrated improvements in surface quality [15] and dimensional accuracy [16], with surface roughness reductions exceeding 40% for AM parts with simple geometries, such as circular or square tube structures. Additionally, AFM has proven effective in finishing the internal channels of complex AM parts, including titanium alloy lattice components featuring arrays of tilted holes [17] and conformal cooling channels in molds [18]. Literature reviews indicate that AFM can effectively remove defect layers, enhance surface quality, and improve the mechanical performance of AM parts.

Despite these advantages, AFM faces challenges in achieving uniform material removal and consistent surface quality after finishing-issues that significantly influence the functionality and inservice performance of the parts. One critical factor in the AFM process is the relationship between the rheological properties of the abrasive media and the surface finishing outcome. Bouland et al. [19] found that material removal in the central regions of V-shaped AM parts is higher after AFM due to uneven flow caused by shear thinning of the abrasive media. Moreover, the viscosity of the abrasive media influences the shear stress exerted on the wall of the workpiece during AFM [20]. Another important consideration is the impact of AM building features on surface finishing. The surface roughness [21] and porosity [22] of AM parts exhibit significant deterioration as the building orientation angle increases. As a result, the non-uniform surface quality and complex structures inherent to AM parts exacerbate the unevenness in the AFM process. Additionally, variations in the angle relationship between the flow direction of abrasive media and the oriented texture of AM parts can alter the surface quality [23].

Researchers have explored various methods to enhance the uniformity of the finishing process in AFM. Some have employed constrained core molds to control velocity distributions [24,25,26], while others have introduced auxiliary physical fields, such as magnetic fields [27,28,29], centrifugal forces [30,31,32], and helical flow channels [33,34], to improve finishing uniformity. Notably, Zhang et al. [35,36] proposed a controllable variant of AFM using magnetic effects to potentially enhance the uniformity of internal channel finishing in AM parts. Nevertheless, AFM continues to suffer from uneven material removal during curved surface polishing, which not only affects the target surface roughness but also impairs dimensional accuracy, particularly in the context of complex channels. During conventional AFM processes, material is gradually removed by abrasive-induced impact and shearing, achieving controllable removal rates in straight or simply shaped channels. However, when applied to geometrically complex regions, especially corners significant limitations emerge. First, due to unstable flow behavior and non-uniform abrasive distribution, localized over- or under-removal often occurs at sharp bends or dead zones, hindering consistent polishing. Second, the presence of high shear stresses and turbulence in corner regions prevents abrasive particles from uniformly contacting the internal wall, especially in narrow or intricately shaped passages. Moreover, the removal efficiency is considerably reduced when processing hard materials (e.g., titanium alloys or Inconel), as abrasive degradation and insufficient material removal become critical issues.

To overcome these limitations, a novel post-processing technique is required to ensure comprehensive and uniform material removal, particularly in complex geometrical features such as corners, bends, and variable-diameter sections. Such a technique must offer precise control over abrasive trajectories and hydrodynamic behavior, thereby mitigating uneven removal and unstable polishing outcomes.

In response, this paper proposes a liquid metal-driven abrasive flow (LM-AF) polishing method tailored for complex AM channels. This approach leverages the high flowability and electric-field responsiveness of liquid metals, combined with abrasive-induced shearing and impact, to improve surface quality in geometrically challenging regions. The key advantages of LM-AF include: Uniform Material Removal: LM-AF establishes a well-distributed flow field within the channel. Through the application of an alternating voltage across the liquid metal droplets placed in the electrolyte at both ends of the flow channel. This electric field induces a flow of liquid metal, which in turn drives the abrasive particles to disperse more uniformly in the geometrically constrained regions, effectively addressing removal inconsistencies in corner regions. Even in geometrically constrained areas, LM-AF generates stable oscillatory flow to ensure consistent polishing performance. Superior Flowability of Liquid Metals: Liquid metals such as Galinstan possess a low melting point (16 °C) and high electrical conductivity, enabling robust flow under low-temperature conditions [37]. Their excellent flow adaptability allows them to access hard to reach areas, such as corners and dead zones, ensuring full-surface polishing, which is difficult to achieve using traditional abrasive media. Precisely Controlled Removal Rates: By regulating the electric field strength and liquid metal flow velocity, abrasive motion can be accurately directed. Electric field-driven motion enhances abrasive–surface interactions, resulting in efficient material removal and finer surface finishes. Strong geometric adaptability: The LM-AF process is highly adaptable to different channel geometries and materials due to its controllable and flow-responsive nature, making it an ideal candidate for polishing AM-fabricated complex channels.

The objective of this study is to address the challenge of non-uniform material removal at internal corners of complex AM channels. A novel polishing technique, driven by liquid metal, is proposed to enhance the uniformity of surface finishing in geometrically complex flow paths.

To evaluate the influence of process variables on surface roughness, an orthogonal experimental design was employed, considering factors such as sodium hydroxide concentration, applied voltage, electric field frequency, and abrasive concentration. This design enables efficient analysis of the individual and interactive effects of each parameter.

Additionally, a hybrid optimization framework integrating Genetic Algorithms (GA) and Neural Networks (NN) was used to refine the polishing parameters. GA was applied to optimize the NN architecture and hyperparameters, while the trained NN model was used in a second-stage GA for parameter optimization. This approach efficiently navigates the high-dimensional process space.

Experimental validation demonstrated significant improvements in surface finish uniformity across different turning angles. The results confirm the reliability and effectiveness of the proposed method, offering a powerful tool for enhancing the surface quality of intricate internal features in AM channels.

## 2. Materials and Methods

### 2.1. Geometric Design of Processing Channels for AFM

To induce turbulent flow within confined channels, four closed flow channel structures with distinct bending angles of 45°, 60°, 90°, and 120° were designed in this study. These geometries were directly constructed using three-dimensional modeling software, with each channel featuring structured internal surfaces tailored to reflect practical polishing conditions. The entire flow domain was pre-defined as a sealed system during the modeling process, eliminating the need for additional assembly or constraint components. The 3D models of channels with different bend angles are shown in Figure 1 and Figure 2. Specifically, Figure 1a–d represent external connector models, whereas Figure 2a–d depict cross-sectional geometry models of channels featuring internal structures.

### 2.2. Turbulent and Multiphase Flow Modeling of Abrasive Flow in Internal Channels

In this study, the slurry flow within the channel is modeled as a homogeneous liquid-solid two-phase flow, in which solid particles are uniformly dispersed within the liquid medium. The solid phase is assumed to consist of spherical particles with an average diameter of approximately 61 µm. It is further assumed that there is no relative slip between the solid and liquid phases, and local equilibrium is maintained over short spatial scales.

The standard k−ε turbulence model, originally proposed by Launder et al. [38], is one of the most widely adopted two-equation models for simulating turbulent flows. However, it has been noted that under conditions of high mean strain rates, the standard k−ε model may yield non-physical results, such as negative normal stresses. To ensure consistency with the physical behavior of turbulent flows, a mathematical constraint on the normal stress is required. The realizable k−ε model was developed specifically to address this limitation by treating the model coefficient $C_{\mu }$ in the eddy viscosity formulation as a function of the mean strain rate. In this model, the turbulent kinetic energy k and its dissipation rate ε are governed by the following transport equations:(1)$$\frac{\partial (\rho k)}{\partial t}+\frac{\partial \left(\rho ku_{i}\right)}{\partial x_{i}}=\frac{\partial }{\partial x_{j}}\left[\left(\mu +\frac{\mu_{t}}{\sigma_{k}}\right)\frac{\partial k}{\partial x_{j}}\right]+G_{k}-\rho \varepsilon$$(2)$$\frac{\partial (\rho \varepsilon )}{\partial t}+\frac{\partial \left(\rho \varepsilon u_{i}\right)}{\partial x_{i}}=\frac{\partial }{\partial x_{j}}\left[\left(\mu +\frac{\mu_{t}}{\sigma_{\varepsilon }}\right)\frac{\partial \varepsilon }{\partial x_{j}}\right]+C_{1}E\rho \varepsilon -C_{2}\rho \frac{\varepsilon^{2}}{k+{\left(\nu \varepsilon \right)}^{\frac{1}{2}}}$$i,j=1,2,3

In the above equations, t represents time and ρ denotes the fluid density. The coordinates ($x_{1},x_{2},x_{3}$) correspond to the Cartesian coordinate axes, i.e., $x_{1}=x,x_{2}=y,x_{3}=z$, while $u_{i}$ denotes the components of the velocity vector u along these directions. The term $G_{k}$ represents the production of turbulent kinetic energy k due to the mean velocity gradients. $\mu_{t}$ is the turbulent (eddy) viscosity, and μ is the dynamic viscosity of the fluid. The strain rate magnitude is denoted by E, and ν is the kinematic viscosity. The turbulent Prandtl numbers for k and ε are $\sigma_{k}=1.0$ and $\sigma_{\varepsilon }=1.2$, respectively. $C_{1}$ and $C_{2}$ are empirical model constants, with $C_{2}$ typically set to 1.9.(3)$$C_{1}=\max\left(0.43,\frac{\eta }{\eta +5}\right)$$(4)$$\eta ={\left(2E_{ij}E_{ij}\right)}^{\frac{1}{2}}\frac{k}{\varepsilon }$$(5)$$E_{ij}=\frac{1}{2}\left(\frac{\partial u_{i}}{\partial x_{j}}+\frac{\partial u_{j}}{\partial x_{i}}\right)$$(6)$$G_{k}=\mu_{t}\left(\frac{\partial u_{i}}{\partial x_{3}}+\frac{\partial u_{3}}{\partial x_{i}}\right)\frac{\partial u_{i}}{\partial x_{3}}$$(7)$$\mu_{t}=\rho C_{\mu }\frac{k^{2}}{\varepsilon }$$(8)$$C_{\mu }=\frac{1}{A_{0}+A_{S}U^{*}\frac{k}{\varepsilon }}$$

In Equation (8), the empirical model constant $A_{0}$ is set to 4.0, and the expressions for calculating $A_{S}$ and $U^{*}$ are given as follows:(9)$$\left.\begin{array}{l} A_{S}=\sqrt{6}\cos\phi \\ \phi =\frac{1}{3}\arccos(\sqrt{6}W)\\ W=\frac{E_{ij}E_{jk}E_{ki}}{{\left(E_{ij}E_{ij}\right)}^{1/2}}\\ U^{*}=\sqrt{E_{ij}E_{ij}+\Omega_{ij}\Omega_{ij}} \end{array}\right\}$$

In these expressions, $\Omega_{ij}$ denotes the components of the rotation rate tensor.

### 2.3. Numerical Simulation of Conventional Abrasive Flow in Internal Channels

A closed-channel geometry was integrally constructed using a 3D modeling software to represent the internal flow domain. A three-dimensional solid–liquid two-phase abrasive flow model was subsequently developed based on this geometry. The computational domain was discretized using tetrahedral mesh elements to accurately capture the complex internal flow behavior, as illustrated in Figure 3.

Prior to initiating the numerical simulations, the initial and boundary conditions for the solid-liquid two-phase abrasive flow within the channel were carefully defined. A fluid flow-particle tracing interface was employed to simulate the trajectories of abrasive particles under the combined influence of drag force and gravity. In the particle tracing module, abrasive particles were introduced from the channel inlet at time zero. The solver iteratively computed the internal distributions of pressure, velocity, turbulent kinetic energy, and particle trajectories. The primary simulation parameters are summarized in Table 1.

The samples used for treatment consist of channels with different angles, manufactured using the FDM (Fused Deposition Modeling) method with the Topobo A1 Mini printer and the Topobo Bambu Studio slicing software (version 2.2.0.65). These samples have dimensions of 40 mm in length, 5 mm in width, and a wall thickness of 2 mm. The internal channel has a length of 40 mm and a width of 5 mm. The samples were printed along the Z-axis using parameters that include a layer height of 0.2 mm and a $45^{\circ }$ printing path angle relative to the layer direction.

In the FDM process, the printing occurs through the layer-by-layer extrusion of heated plastic filament, where the printer head melts the filament and extrudes it through a hot nozzle, building up the sample layer by layer. The internal channel surface of the samples is related to the building direction angle (θ), which is the angle between the building direction and the surface’s outer normal. For the given channels, the angles can be categorized as follows: $45^{\circ }$ channel surface (U), with a building orientation angle of $45^{\circ }$; $\;60^{\circ }$ channel surface (D), with a building orientation angle of $60^{\circ }$; $90^{\circ }$ channel surface (V), with a building orientation angle of $90^{\circ }$; $120^{\circ }$ channel surface (V), with a building orientation angle of $120^{\circ }$.

When manufacturing samples with varying channel angles $\left(45^{\circ },60^{\circ },90^{\circ }\right.$, and $\left.120^{\circ }\right)$ using FDM, the channel angles directly influence the printing path and layer adhesion, which in turn affects the surface quality.

### 2.4. Polishing Experiment of Channel Inner Walls Using Conventional Solid–Liquid Two-Phase Abrasive Flow

To validate the surface finishing performance of the proposed solid–liquid two-phase abrasive flow at channel corner regions with bending angles of 45°, 60°, 90°, and 120°, polishing experiments were conducted based on the simulation results of the four flow path configurations. The experimental system was designed and assembled as illustrated in Figure 4. The test platform consisted of an air compressor, air treatment unit, solenoid valve, pneumatic cylinder, controller, and 3D-printed channels fabricated via additive manufacturing. The workpieces were PETG (polyethylene terephthalate glycol-modified) curved flow channels with internal angles of 45°, 60°, 90°, and 120°, respectively. The workpiece was sealed at both ends using pistons, which were driven by compressed air through a solenoid valve. This setup enabled the abrasive medium to reciprocate through the internal channel under pneumatic actuation, thereby realizing the polishing process along the inner wall.

To ensure consistency between the experimental conditions and simulation results, the volume ratio of abrasives to liquid phase in the polishing medium was kept identical to that used in the simulations. To minimize temperature-induced variability and facilitate real-time observation of surface morphology changes during polishing, the experiment was conducted under a controlled reciprocating regime. A total of 7200 polishing cycles (each cycle defined as one reciprocating motion of the abrasive slurry through the channel) was performed. Each cycle lasted for 4 s, consisting of 1 s of forward motion, a 3 s pause, and then reverse motion. This timing scheme helped maintain thermal stability during processing while enabling periodic surface roughness measurements.

To achieve a stable inlet velocity of 2 m/s during the polishing process, a precision pressure regulator on the air treatment unit was employed to control the piston speed of the pneumatic cylinder. Based on the principle of flow continuity, the average velocity of the piston was maintained at approximately 75 mm/s. The remaining processing parameters are summarized in Table 2.

Prior to the polishing experiments, the inner surface of the additively manufactured channel exhibited considerable initial roughness due to the inherent characteristics of the printing process. To establish a consistent baseline, the workpiece was pre-polished using sandpaper, and the initial surface roughness was measured to be approximately Sa 62 µm, as shown in Figure 5. For subsequent analysis, the corner region of the channel was uniformly divided into four subregions, labeled a through d.

### 2.5. Mechanism of Liquid Metal-Driven Abrasive Flow Under Alternating Electric Field

As reported in reference [39], when an alternating voltage is applied at both ends of the liquid metal, bipolar electro-shear stress induces a strong quadrupolar vortex field in the transverse direction, driven by the applied sinusoidal waveform. This stress is proportional to the square of the electric field intensity, as illustrated in Figure 6. Furthermore, reference [40] demonstrated that liquid metal-driven abrasive flow can effectively remove material from the internal surfaces of complex additively manufactured flow channels, offering a promising solution for post-processing in such intricate geometries.

Inspired by the aforementioned studies, the generation of vortical flow induced by liquid metal under an alternating electric field can be harnessed to drive abrasive particles for localized material removal in corner regions. In this study, following conventional abrasive flow machining (AFM), a liquid metal-driven abrasive flow polishing technique was employed to further process the corner regions of the flow channels. Both numerical simulations and experimental investigations were conducted to evaluate the effectiveness of this approach. Figure 7 illustrates the schematic diagram of localized material removal induced by liquid metal-driven abrasive flow under an alternating electric field. A liquid metal droplet (EGaln, composed of 90 wt% gallium and 10 wt% indium) is injected into a sealed microchannel partially filled with an aqueous NaOH solution, in which boron carbide $\left(B_{4}C\right)$ abrasive particles are uniformly dispersed. Two titanium electrodes are inserted at both ends of the channel, and an AC/DC programmable power supply is applied to generate an alternating electric field. Driven by bipolar electrowetting-induced shear forces, the liquid metal droplet excites a quadrupolar vortex flow in the electrolyte, which enhances the localized motion of abrasive particles around the corner region. This mechanism enables directional material removal in geometrically confined zones with high precision. The detailed simulation parameters are summarized in Table 3.

### 2.6. Surface Roughness Measurement

To ensure the accuracy and consistency of surface quality assessments, surface roughness was measured using an ultra-depth-of-field microscope (Model: Keyence VHX-7000, Osaka, Japan). The microscope was calibrated prior to use to guarantee precision in all measurements. For each experimental sample, surface roughness was quantified by measuring the Sa parameter, which represents the arithmetic average roughness. The measurement range of the microscope covered up to 200 mm×200 mm, with a vertical resolution of 0.01 μm to ensure fine detail capture, especially for complex geometries such as internal corners.

Additionally, to minimize surface measurement errors caused by rough data, a Gaussian filter with a cutoff wavelength of 0.8 mm was applied to the raw data. This filtering method effectively eliminates high-frequency noise, ensuring that only the relevant surface features are analyzed. The measurements were taken at multiple positions within each region of the channel (i.e., regions a, b, c, and d for each flow path) to provide a comprehensive representation of the surface quality at different locations within the polished areas.

This measurement protocol ensures reliable and reproducible roughness values, contributing to the validation of the proposed LM-AF method’s effectiveness in improving surface finish, especially at geometrically complex internal corners.

## 3. Results and Discussion

### 3.1. Numerical Simulation Analysis

Figure 8 illustrates the pressure distribution of a solid–liquid two-phase abrasive flow in channels with four typical turning angles (45°, 60°, 90°, and 120°), aiming to reveal the influence of geometric configuration on local pressure gradients and material removal behavior. The results indicate that all turning angles exhibit a marked pressure drop, transitioning from positive to negative pressure zones. The resulting large local pressure differentials and vortex formations are critical factors affecting abrasive impact intensity and trajectory distribution.

In the 45° sharp-angled channel (Figure 8a), the peak pressure reaches approximately $1.18\times 10^{4}\;Pa$, while the minimum pressure in the negative zone is as low as −$8.7\times 10^{3}\;Pa$, indicating a steep pressure gradient. The high-speed flow tends to adhere to the outer wall, intensifying particle impingement in that region. Conversely, the low-pressure inner wall region is prone to backflow and abrasive accumulation, leading to insufficient material removal.

For the $60^{\circ }$ angled structure (Figure 8b), although the pressure variation is slightly alleviated, a noticeable asymmetry remains at the corner. The maximum and minimum pressures are $1.1\times 10^{4}\;Pa$ and $-8.2\times 10^{3}\;Pa$, respectively. The enhanced outer wall impingement is evident, while the central vortex zone may still obstruct particle migration to the inner corner, resulting in potential polishing dead zones.

In the 90° right-angled channel (Figure 8c), the most significant pressure gradient is observed, with a peak pressure of $2.12\times 10^{4}\;Pa$ and a minimum of $-1.92\times 10^{4}\;Pa$. The abrupt flow turning induces strong vortex deviation, directing particles toward the outer wall. This results in highly uneven material removal-insufficient impingement on the inner wall, high impact on the outer region, and a typical “jet concentration” effect due to abrasive particle aggregation.

In contrast, the $120^{\circ }$ obtuse-angled channel (Figure 8d) exhibits the smoothest pressure transition, with positive and negative pressures of $7.4\times 10^{3}\;Pa$ and $-7.8\times 10^{3}\;Pa$, respectively. The reduced pressure gradient and minimized negative-pressure zone enhance flow stability, weaken jet impact fluctuations, and promote more uniform abrasive particle distribution across both inner and outer walls-contributing to a more consistent polishing performance.

Figure 9 presents the velocity distributions of solid–liquid two-phase abrasive flow in channels with four typical turning angles (45°, 60°, 90°, and 120°), aiming to evaluate the influence of geometric turning features on local flow behavior and material removal uniformity.

Overall, the results show that with increasing turning angle, the velocity distribution becomes progressively more uniform, with reduced main flow deflection and diminished velocity differentials between the inner and outer walls-conditions favorable for achieving uniform abrasive impingement.

In the 45° sharp-angled channel (Figure 9a), the fluid experiences severe directional deviation and sharp curvature at the turning point, producing a distinct high-velocity jet along the outer wall (peak velocity approximately 3.6 m/s). Conversely, a large low-velocity zone forms on the inner wall, often accompanied by backflow phenomena. These effects can lead to particle accumulation and insufficient inner-wall impingement, resulting in markedly uneven material removal.

In contrast, the 60° oblique-angled channel (Figure 9b) shows a milder turning profile. The high velocity region expands toward the center of the channel, while the low-velocity zone shrinks. Local recirculation is suppressed, and abrasive coverage on the inner wall improves. This configuration shows enhanced uniformity in material removal in the corner region.

In the 90° right-angled channel (Figure 9c), the flow undergoes a sharp turn over the shortest path, causing the inertial flow to adhere tightly to the outer wall, forming a pronounced high-speed jet (peak velocity ∼3.09 m/s). The velocity near the inner wall drops significantly, in some regions approaching stagnation. This creates a highly asymmetric velocity field, characterized by intense outer-wall impingement and the formation of dead zones along the inner wall-typical of the “strong outside, weak inside” polishing pattern.

By comparison, the 120° obtuse-angled channel (Figure 9d) exhibits smoother directional transition and gentler velocity variations, leading to a more stable flow structure. The high-velocity regions exhibit symmetric jet profiles along the channel walls, with a maximum velocity of approximately 2.76 m/s. The similar velocity distribution between the inner and outer walls enhances the uniformity of abrasive contact and overall polishing consistency.

Figure 10 illustrates the distribution of turbulent kinetic energy (TKE) of solid–liquid two-phase abrasive flow within channels featuring four typical turning angles (45°, 60°, 90°, and 120°), aiming to reveal the influence of local velocity fluctuations and shear disturbance levels on abrasive impact potential and material removal uniformity.

TKE reflects the intensity of local velocity fluctuations and shear agitation, serving as a key indicator for evaluating material removal potential and uniformity.

In the 45° sharp-angled configuration (Figure 10a), TKE exhibits a pronounced increase on the outer corner, with a peak value of $0.71{\;m}^{2}/s^{2}$, indicating intense local shear and flow deviation. This promotes the concentration of abrasive particles and their impingement on the outer wall. However, the TKE in the inner region remains low, leading to insufficient abrasive agitation and an evident “strong outside, weak inside” removal pattern, indicative of poor uniformity.

In the 60° angled structure (Figure 10b), although the asymmetric distribution with higher TKE on the outer side persists, the intensity gradient is more moderate than that of the 45° case. The high TKE zone contracts toward the central streamline, and the shear gradient is smoother, allowing abrasives to receive more balanced kinetic input across the flow field. This contributes to improved uniformity in the impingement pattern at the corner region.

In the 90° right-angled configuration (Figure 10c), TKE reaches the highest peak of all cases, up to $1.27{\;m}^{2}/s^{2}$, with turbulence concentrated on the outer wall of the corner. The high shear vortex enhances localized abrasive impingement, facilitating efficient removal. However, the accompanying strong deviation flow results in severe agitation loss near the inner wall, compromising processing consistency across the channel cross-section.

In contrast, the 120° obtuse-angled configuration (Figure 10d) exhibits a more evenly distributed TKE field, with a peak value of approximately $0.62{\;m}^{2}/s^{2}$. The disturbance region extends across both inner and outer walls, and the shear transition is smoother. Abrasive motion is more continuous and evenly distributed, favoring uniform material removal in the complex geometry of the turning region.

Figure 11 presents the particle trajectory distributions of solid–liquid two-phase abrasive flow under different turning angles (45°, 60°, 90°, and 120°), aiming to evaluate how corner geometry affects flow continuity, particle migration paths, and erosion uniformity.

The overall trend reveals that as the turning angle increases, the internal flow becomes more stable, abrasive particles distribute more uniformly, and both surface coverage and material removal consistency improve markedly.

In the 45° sharp-angled configuration (Figure 11a), abrupt flow deflection leads to noticeable particle accumulation, deviation, and detachment. Some particles recirculate or stagnate near the inner corner, forming low-density zones and erosion blind spots, which significantly hinder the effective removal of material from the inner wall.

In the 60° configuration (Figure 11b), particle trajectories become more continuous, and the wall adhering flow improves, enabling a more balanced abrasive impact on both the inner and outer surfaces. However, due to the persistent effect of flow inertia, localized particle-depleted areas still appear near the inner wall, suggesting that although erosion uniformity improves, it remains influenced by boundary-induced disturbances.

The 90° right-angle configuration (Figure 11c) exhibits a pronounced flow direction change, with the abrasive stream strongly biased toward the outer wall, forming a dense, high-speed erosion zone. In contrast, the inner region shows sparse abrasive coverage and visible voids, presenting the typical feature of “strong outer, weak inner” erosion and the most significant inconsistency in material removal among the tested geometries.

In the 120° obtuse-angle configuration (Figure 11d), a more uniform particle distribution is observed. Abrasive trajectories remain stable and evenly spread along both the inner and outer surfaces, with no evident recirculation or accumulation. This geometry helps maintain flow continuity, suppress inertial deviation, and enhance the uniformity and consistency of wall erosion.

Figure 12, Figure 13, Figure 14 and Figure 15 compare the surface roughness values (Sa) across different regions of flow channels with corner angles of 45°, 60°, 90°, and 120°, respectively, after polishing via conventional abrasive flow machining (AFM). The results demonstrate that traditional AFM significantly improves the initial surface roughness of all channel geometries. However, for all corner configurations, region d (the turning corner area) consistently exhibits higher residual roughness compared to regions a, b, and c, indicating inferior material removal in the corner zones. This observation is in good agreement with the simulation results, suggesting that conventional AFM suffers from flow stagnation and abrasive inaccessibility in corner areas, leading to polishing dead zones.

To quantitatively evaluate this effect, surface roughness in each region was measured using a 3D surface profilometer (UNI-TUTG932E/962E, China). It was found that for the 45°, 60°, and 90° channels, region d consistently exhibited significantly higher Sa values than Regions a, b, and c. In contrast, the 120° obtuse-angle channel showed relatively uniform Sa values across all regions. This further confirms that sharper turning angles are more prone to uneven material removal and poor abrasive coverage.

Therefore, subsequent studies should focus on enhancing the polishing effectiveness specifically in region d to achieve uniform surface quality across the entire inner channel surface.

### 3.2. Evaluation of Simulated Flow and Particle Behavior

Figure 16 and Figure 17 illustrate the locally enhanced flow dynamics and particle motion within a confined microchannel under the simulation conditions summarized in the preceding table. As shown in the magnified view of Figure 16, the injected liquid metal droplet induces a characteristic annular vortex structure, resulting in a pronounced increase in local flow velocity near the channel corner. Figure 17 presents a magnified view of the particle trajectory distribution, revealing a significant redistribution of abrasive motion. Specifically, the mobility and impact density of particles are substantially enhanced in the corner region, effectively improving the local material removal capability in geometrically constrained zones and achieving directional and uniform micro-scale polishing. These findings provide not only a physical insight into the localized enhancement mechanism but also a theoretical foundation for subsequent targeted reinforcement processing in corner region d, addressing the deficiencies observed in traditional abrasive flow machining.

### 3.3. Parametric Study on Single-Factor Polishing Experiments in a 90° Channel Corner

Given the aforementioned findings, it is clear that in the liquid metal-driven abrasive flow polishing of 90° channel corners, critical process parameters-including NaOH concentration, applied voltage amplitude, AC frequency, and abrasive concentration have a significant impact on polishing performance. To elucidate the influence of these variables on surface roughness, a series of single factor experiments were conducted for liquid metal-driven abrasive flow polishing in 90° corner geometries. Table 4 presents the detailed design of the single-factor experiments.

The experimental setup comprised five levels for each of the four parameters: NaOH concentration (0.2, 0.4, 0.6, 0.8, and 1.0 mol/L), voltage amplitude (10, 20, 30, 40, and 50 V), abrasive mass fraction (10%, 15%, 20%, 25%, and 30%), and AC frequency. The polishing time for each experiment was fixed at 12 h, and the average initial surface roughness of the samples was approximately 62 µm.

To further optimize the surface removal performance of liquid metal-driven abrasive flow machining (LM-AF) in the corner regions of complex flow channels, this study focuses on a 90° right-angle channel and systematically investigates the effects of NaOH electrolyte concentration, electric field strength, abrasive mass fraction, and AC field frequency on the surface roughness (Sa) of corner regions through a series of single-factor experiments. Figure 18, Figure 19, Figure 20 and Figure 21 illustrate the trends in surface roughness variation before and after machining under different parameter conditions.

As shown in Figure 18, increasing NaOH concentration significantly influences surface roughness. With the NaOH concentration raised from 0.2 mol/L to 1.0 mol/L, the surface roughness in the corner region gradually decreases from 9.76 µm to 8.78 µm, representing a reduction of 10.2%. This improvement is attributed to the enhanced electrochemical reactivity and flowability of the liquid metal in a highly alkaline environment, resulting in stronger electric-field-induced propulsion of the abrasive media. Consequently, the abrasive particles exhibit improved mobility and impact efficiency within the channel. Furthermore, the increased ionic strength of the solution promotes stronger shear-induced turbulence under the applied electric field, facilitating uniform distribution and effective contact of the abrasives with the irregular surfaces of the corner regions.

Figure 19 illustrates the influence of electric field intensity on surface roughness. As the applied voltage increases from 10 V to 50 V, the surface roughness significantly decreases from 7.9 µm to 6.87 µm. This trend highlights that electric field strength is a critical parameter in regulating particle mobility in the LM-AF system. A stronger electric field enhances the electrohydrodynamic force exerted on the liquid metal, which in turn improves the kinetic energy of the abrasive particles. This increased energy facilitates more effective micro-scale impact and shearing against the wall surface, thereby improving material removal efficiency. Furthermore, the intensified electric field mitigates particle stagnation and accumulation near the corner region by disrupting low-velocity flow zones, ultimately enhancing polishing uniformity at the turning areas of complex internal channels.

Figure 20 illustrates the influence of abrasive mass fraction on surface roughness. The results reveal that as the abrasive concentration increases from 10% to 25%, the surface roughness (Sa) decreases progressively, reaching a minimum value of 7.96 µm at 20%. However, a slight increase in roughness is observed when the concentration reaches 30%. This trend suggests that an appropriate abrasive concentration is beneficial for establishing a stable shear layer, thereby enhancing the frequency and uniformity of abrasive impacts and improving the overall surface finish. Conversely, excessively high concentrations may lead to particle agglomeration or hindered flow uniformity, which reduces local flow dynamics and impact activity, ultimately resulting in diminished polishing efficiency.

Figure 21 presents the effect of alternating current (AC) field frequency on polishing performance. The results indicate that the surface roughness reaches a minimum of 6.88 μm at 100 Hz. However, a slight increase in roughness is observed around 300 Hz, after which the variation tends to stabilize under higher frequency conditions. This suggests that an appropriate frequency facilitates the formation of a relatively stable flow field with sufficient driving force, promoting continuous wall-adhering motion and stable impingement of abrasive particles. In contrast, excessively high frequencies result in shortened electrical response cycles, which hinder the formation of effective shear flow and reduce the overall material removal efficiency.

A comprehensive analysis of the preliminary single-factor experiments demonstrated that NaOH concentration, electric field strength, abrasive concentration, and AC frequency each have a significant impact on material removal behavior at the inner corners of the channel. These findings serve as an initial validation of the process feasibility of liquid metal-driven abrasive flow (LM-AF) polishing for internal surface finishing in complex additively manufactured channels, and they reveal fundamental trends and engineering characteristics associated with parameter control.

Nevertheless, although the single-factor approach is effective in identifying the individual influence of each variable, it is inherently limited in its ability to capture the interactions among multiple parameters. As such, the optimal parameter set obtained at this stage may not represent the global optimum. To address these limitations and further enhance the robustness and overall performance of the polishing process, future work will involve a systematic optimization of the process parameter space through orthogonal experimental design combined with a GA-NN-GA (Genetic Algorithm Neural Network-Genetic Algorithm) hybrid optimization strategy. This approach aims to achieve more efficient and uniform surface treatment of geometrically complex internal structures.

### 3.4. Orthogonal Polishing Experiments for the 90° Channel Corner

Although single-factor experiments can clearly reveal the impact of each factor on material surface quality, they do not provide an assessment of the relative contributions and significance of each factor to the target metric. To further investigate the abrasive flow polishing process of the $90^{\circ }$ corner geometry driven by liquid metal and optimize surface processing quality, an orthogonal experimental design is adopted in this study, with the surface roughness at the channel bottom as the performance evaluation indicator. The orthogonal experimental method efficiently analyzes the combined effects of multiple factors and their interactions on roughness by systematically arranging experiments. Compared to single-factor experiments, the orthogonal method allows for the simultaneous examination of multiple factors and provides a more accurate identification of the significance of each factor on surface quality. This approach offers scientific evidence and optimization guidance for the efficient application of the liquid metal-driven abrasive flow polishing process. The factors and their levels selected for the orthogonal experiment of abrasive flow polishing in the 90° corner geometry driven by liquid metal are shown in Table 5.

The standard orthogonal table L25(5^4^) for the orthogonal experimental arrangement is shown in Table 6. The four factors with five levels are as follows:

Each row in Table 7 represents a distinct combination of processing parameters. The experimental number 3 is denoted as A_1_B_3_C_3_D_3_, where the subscripts correspond to the level numbers of each factor. Therefore, experiment 3 corresponds to the following conditions: NaOH concentration of 0.2 mol/L, electric field intensity of 30 V, AC field frequency of 300 Hz, and abrasive mass fraction of 20%. The same logic applies to the other experiments. The required workpieces for the 25 experimental conditions are shown in the figure below. Each workpiece has dimensions of 40 mm× 5 mm×2 mm, and is made of 3D-printed PETG. The polishing time for each group is set to 8 h. For each experiment, three measurements were taken for each condition, and the average value of these measurements was used in the analysis. The experimental results of all 25 groups are summarized in Table 7.

This study proposes an optimization method that combines Genetic Algorithm (GA) and Neural Network (NN), referred to as GA-NN-GA, for optimizing the surface quality in the Liquid metal driven Abrasive flow (LM-AF) process, specifically focusing on the surface roughness at the 90° corner of the flow channel. LM-AF is an advanced precision machining technique, where multiple processing parameters significantly impact the surface quality. Traditional optimization methods often encounter challenges related to accuracy and computational efficiency when addressing the complex interactions between multiple parameters. To overcome these issues, the GA-NN-GA method is introduced, leveraging the global optimization capability of GA and the high-precision surface roughness prediction ability of NN, thus providing a robust solution for surface quality optimization in the LM-AF process.

The first phase of the optimization process involves acquiring data from an L25 orthogonal experimental design that considers four key process parameters: NaOH concentration (mol/L), voltage (V), frequency (Hz), and abrasive mass fraction (%). Each experimental combination is associated with a corresponding surface roughness value (Sa), which serves as the target for the NN model. A neural network is then trained using this data to predict Sa based on the input parameters. The architecture of the NN includes an input layer with four variables, a hidden layer with 10 neurons, and an output layer predicting surface roughness. During training, backpropagation is used to minimize the Mean Squared Error (MSE) between predicted and actual Sa values. The prediction capacity of GA-NN-GA is evaluated in this study using the coefficient of determination ($R^{2}$), mean absolute error (MAE), and root mean square error (RMSE). The evaluation formulas are displayed as follows in Equations (10)–(12):(10)$$R^{2}=1-\frac{\sum_{i}\;\;{\left({\hat{y}}^{\left(i\right)}-y^{\left(i\right)}\right)}^{2}}{\sum_{i}\;\;{\left(\bar{y}-y^{\left(i\right)}\right)}^{2}}$$(11)$$MAE=\frac{1}{N}\sum_{i=1}^{N}\;\;\left|y_{i}-{\hat{y}}_{i}\right|$$(12)$$RMSE=\sqrt{\frac{1}{N}\sum_{i=1}^{N}\;\;{\left(y_{i}-{\hat{y}}_{i}\right)}^{2}}$$
where $y_{i}$ stands for the true value, ${\hat{y}}_{i}$ for the predicted value, and $\bar{y}$ for the sample mean. N denotes the total number of samples. $R^{2}$ quantifies the degree to which the model fits the data when assessing machine learning performance; $R^{2}\leq 1$. The machine learning model performs better in terms of prediction the closer $R^{2}$ is to 1. The projected value is closer to the true value when the deviation between the two is smaller, as measured by the root mean square error (RMSE). The more accurate the forecast, the less the error, which is determined by the mean absolute error (MAE) between the true and projected values.

To assess the model’s performance, three key statistical metrics were utilized: $R^{2}$, MAE, and RMSE. As shown in Table 8, the $R^{2}$ value of 0.99642 indicates that the model explains 99.64% of the variance in the data, demonstrating a highly accurate fit and strong predictive capability. The MAE of 0.335895 reflects the average absolute deviation between predicted and actual values, signaling minimal prediction errors and suggesting reliable model accuracy. Additionally, the RMSE value of 0.46282, which measures the standard deviation of the prediction errors, indicates that the model’s errors remain within an acceptable range, further confirming its robustness and consistency. Collectively, these metrics substantiate the model’s effectiveness in predicting surface roughness and its suitability for optimization tasks.

Once the NN model is trained, GA is employed to optimize the process parameters. Each combination of parameters is encoded into a chromosome, and GA works by evaluating the fitness of each combination based on its predicted Sa value. The optimization process includes the selection of the best-performing individuals, crossover, and mutation operations to evolve new generations of solutions. The optimization terminates once the maximum number of generations is reached, or further improvements in fitness become negligible. Figure 22 illustrates the comprehensive flow of the GA-NN-GA optimization method, which is employed for the surface quality optimization of the LM-AF process.

Through the GA-NN-GA methodology, iterative optimization of the process parameters is achieved. In each generation, the neural network predicts Sa for various parameter combinations, and these predictions serve as the fitness function for the GA. The GA continues to identify the parameter combination that minimizes the predicted Sa, eventually determining the optimal process parameters. These optimal parameters include NaOH concentration (0.99998), voltage (49.993 V), frequency (428.27 Hz), and abrasive mass fraction (10.085%), which together minimize surface roughness, with the final predicted Sa value of 7.8201 μm. In practical applications, the optimization results can be appropriately rounded based on the precision of the equipment, process requirements, and the adjustment range. For the parameters considered (NaOH concentration, voltage, frequency, and abrasive mass fraction), these values can be rounded to a reasonable level of precision. The final optimized process parameters are NaOH concentration (1), frequency (428.3 Hz), voltage (50 V), and abrasive mass fraction (10%).

Figure 23 represents the optimization process was validated through the GA convergence curve, which showed rapid convergence. After approximately 100 generations, the MSE dropped significantly from 0.14 to 0.06, stabilizing at this level, indicating the strong optimization ability of GA. Furthermore, the predicted vs. actual comparison plot demonstrated excellent agreement between the predicted surface roughness values and the experimental measurements. This validated the model’s accuracy and highlighted the potential of the GA-NN-GA optimization method for surface roughness prediction and control. Robustness analysis was performed to assess the reliability of the optimized model, showing that the method consistently provided stable process parameter combinations with minimal error. This indicates that the GA-NN-GA approach significantly improves the precision and reliability of the manufacturing process.

### 3.5. Liquid Metal-Driven Abrasive Flow Polishing of Internal Channel with Different Bends

In the initially polished channels subjected to conventional abrasive flow machining (AFM), it was observed that the surface roughness (Sa) in the corner region (denoted as region d) of the 45°, 60°, and 90° channels was significantly higher than that in adjacent straight segments (a, b, and c), indicating notable material removal nonuniformity. In contrast, the 120° channel exhibited approximately uniform roughness values across all measured regions, suggesting a more homogeneous polishing performance.

To address the localized roughness disparity in the sharper-angled channels, localized polishing was further applied to the corner region (d) using a liquid metal-driven abrasive flow under an alternating current (AC) electric field. The aim was to achieve a uniform Sa value across the entire channel surface, particularly aligning the roughness of the corner region with that of the straight segments.

The processing parameters were selected based on the optimal conditions for the 90° channel, which were obtained through the GA-NN-GA optimization algorithm. Specifically, the 90° channel was processed under the following conditions: NaOH concentration of 1 mol/L, electric field intensity of 50 V, boron carbide abrasive (625 mesh) at a mass fraction of 10%, AC frequency of 428.3 Hz, liquid metal dosage of 0.3 g, and a total polishing time of 12 h.

For the 60° and 45° channels, the treatment duration was extended to account for the decreased polishing efficiency in sharper turns. The processing time was increased to 36 h for the 60° channel and 74 h for the 45° channel, respectively. It is worth noting that additional factors such as the electric field intensity variation across different channel geometries, and the distance between the liquid metal medium and the wall surface, may also influence the local material removal rate. However, these effects are beyond the scope of the present study and are not discussed in detail herein.

Figure 24, Figure 25 and Figure 26 illustrate the surface roughness (Sa) of region d in channels with turning angles of 45°, 60°, and 90°, after undergoing secondary polishing using liquid metal-driven abrasive flow (LM-AF). This secondary treatment was applied to compensate for the material removal nonuniformity caused by dead zones in traditional abrasive flow machining (AFM), which were particularly evident in the turning regions. As observed, the surface roughness in region d was significantly reduced following LM-AF processing, compared to the results obtained by traditional AFM alone.

These results validate the capability of the LM-AF technique, driven by alternating electric fields, to overcome the intrinsic limitations of conventional AFM in handling geometrically constrained regions such as sharp turns and internal corners. By leveraging controllable vortices and localized energy input, LM-AF effectively enhances the abrasive particle dynamics in stagnation-prone regions, thereby ensuring more uniform material removal and consistent surface finishing. This outcome demonstrates the great promise of LM-AF in addressing the bottlenecks in postprocessing of additively manufactured internal channels with complex geometries.

In this study, the new method was specifically applied to region d, which typically presents the most challenging polishing conditions. Previous research has indicated that the surface roughness (Sa) in region d of the 45°, 60°, and 90° channel angles was significantly larger compared to regions a, b, and c. In contrast, the 120° channel angle showed similar roughness values across all regions. Therefore, the new polishing method was applied exclusively to region d of the 45°, 60°, and 90° channel angles, while no treatment was applied to the 120° channel angle, where the roughness values in all regions were nearly identical. Table 9 summarizes the surface roughness (Sa) values for each region, comparing the results obtained using the conventional AFM method with those measured using the new method. As shown, the roughness values for regions a, b, and c remained unchanged, as these areas were not treated with the new method. However, for region d, the new method significantly reduced the surface roughness (Sa), demonstrating its effectiveness in enhancing the polishing process in more complex regions.

## 4. Conclusions

This study addresses the challenge of non-uniform polishing in additively manufactured (AM) channels, particularly at internal corners with varying angles. A novel liquid metal-driven abrasive flow (LM-AF) polishing method is proposed, leveraging the unique properties of liquid metals under an alternating electric field. By creating a constrained flow domain with a thin-walled cover structure integrated into the target surface, the LM-AF technique ensures that the abrasive flow is confined within the channel, enhancing material removal in complex geometries.

The corner regions of flow channels with angles of $45^{\circ },60^{\circ },90^{\circ }$, and $120^{\circ }$ were subdivided into distinct zones for detailed analysis. Numerical simulations and experimental results showed that, although traditional abrasive flow machining (AFM) significantly improves the overall surface quality, the corner region (zone d) consistently exhibited higher residual roughness compared to other regions. This indicates the limitations of conventional AFM in achieving consistent polishing in complex geometries, especially at sharp corners. The LM-AF method successfully addressed these challenges by enhancing abrasive particle dynamics and material removal at these critical locations.

Focusing on the $90^{\circ }$ channel, single-factor experiments were conducted to evaluate the influence of key process parameters—such as NaOH concentration, electric field intensity, abrasive mass fraction, and AC frequency—on surface roughness (Sa) in the corner region. Using a hybrid Genetic Algorithm (GA) and Neural Network (NN) optimization framework (GA-NN-GA), surface roughness (Sa) was effectively minimized, with the final predicted value of 7.8201 μm. The optimization process showed a high correlation between predicted and actual Sa values, confirming the reliability and potential of this method in manufacturing process optimization.

Using the optimized parameters, the LM-AF method significantly reduced the surface roughness at internal corners of flow channels with angles of $45^{\circ },60^{\circ }$, and $90^{\circ }$. Specifically, the Sa values were reduced from 25.365 μm to 15.780 μm, from 22.950 μm to 15.718 μm, and from 10.933 μm to 10.055 μm, respectively. These results demonstrate the LM-AF method’s ability to achieve uniform roughness across the internal corners, effectively overcoming the non-uniformity observed with traditional AFM methods.

The proposed LM-AF method offers a reliable solution for improving surface quality in manufacturing processes, particularly in precision machining and polishing of geometrically complex AM channels. However, there are limitations in this study. Future work can explore alternative neural network architectures, more sophisticated optimization schemes, and broader material applications. By expanding the optimization framework to include deeper neural networks, AutoML methods, or new optimization strategies, the robustness and efficiency of the method could be further improved. Additionally, applying this method to a wider range of materials and manufacturing processes will enhance surface quality across diverse industrial applications.

In future research, we plan to explore the impact of the thermal effects induced by electric current on the surface quality, particularly in relation to the conductivity of the medium and its interaction with the flow of liquid metal. This will be integrated into an extended research framework, providing a more comprehensive understanding of the factors influencing the polishing process. The inclusion of these effects will further enhance the optimization of the surface treatment in additive manufacturing applications and open new avenues for improving the uniformity of material removal in complex geometries.

## Figures and Tables

**Figure 1 micromachines-16-00987-f001:**
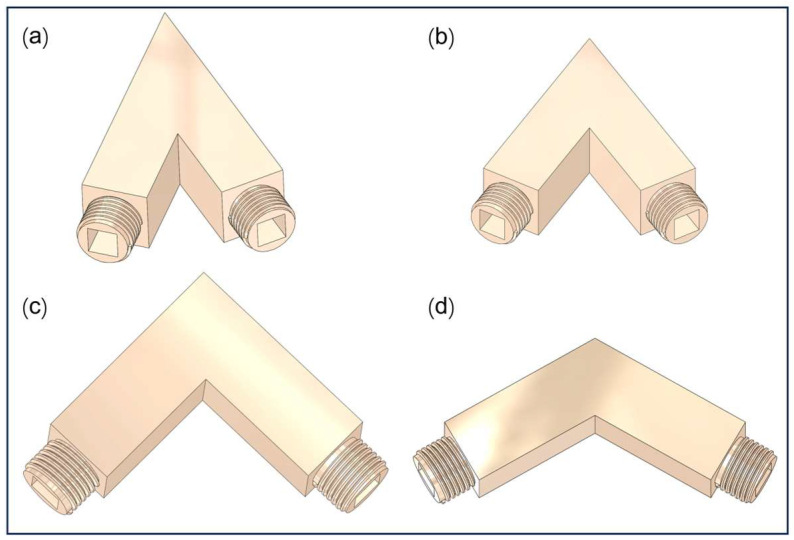
Three-dimensional geometry models of channels. (**a**) 45° bend; (**b**) 60° bend; (**c**) 90° bend; (**d**) 120° bend.

**Figure 2 micromachines-16-00987-f002:**
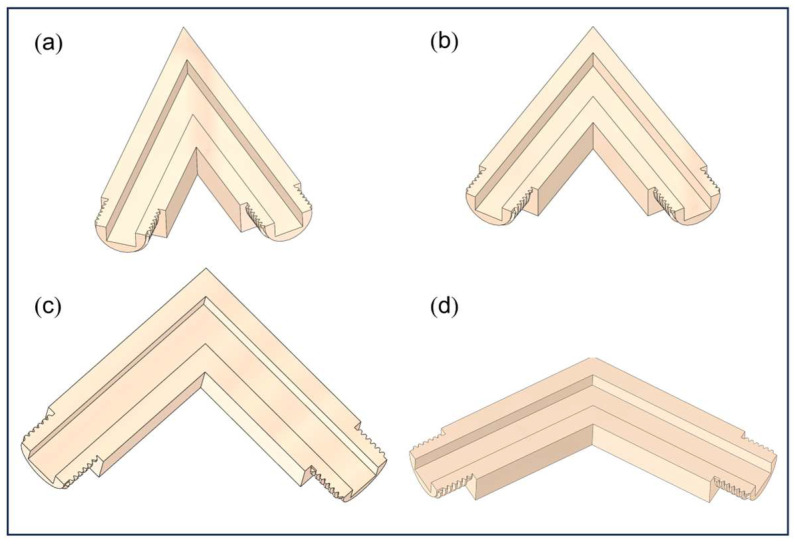
Three-dimensional cross-sectional geometry models of channels with internal structures. (**a**) 45° bend; (**b**) 60° bend; (**c**) 90° bend; (**d**) 120° bend.

**Figure 3 micromachines-16-00987-f003:**
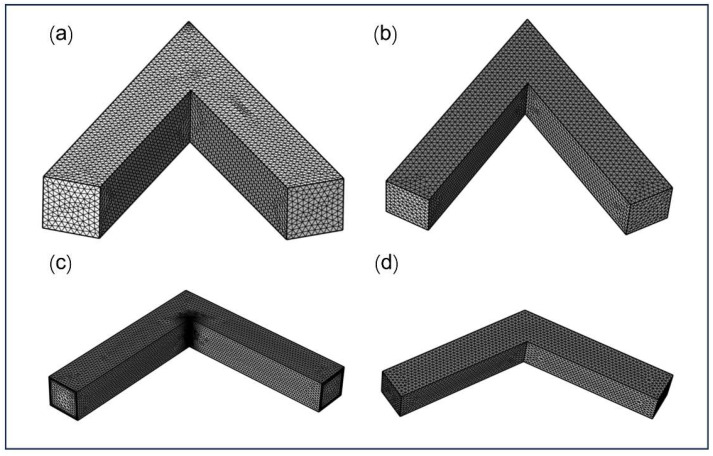
Computational mesh of the liquid flow domain in channels with different turning angles: (**a**) $45^{\circ }$, (**b**) $60^{\circ }$, (**c**) $90^{\circ }$, and (**d**) $120^{\circ }$.

**Figure 4 micromachines-16-00987-f004:**
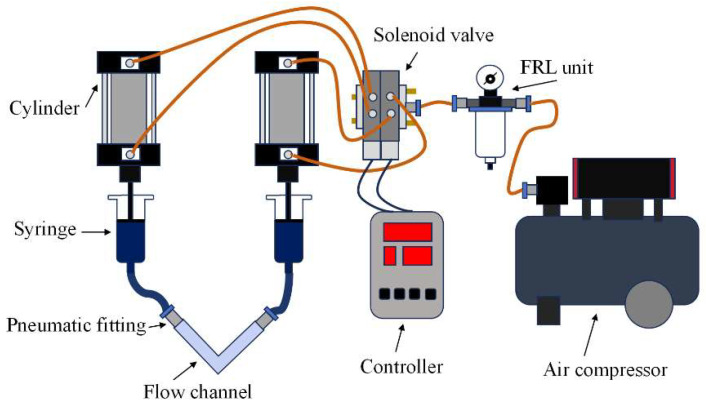
Schematic diagram of the solid–liquid two-phase abrasive flow machining (AFM) experimental setup.

**Figure 5 micromachines-16-00987-f005:**
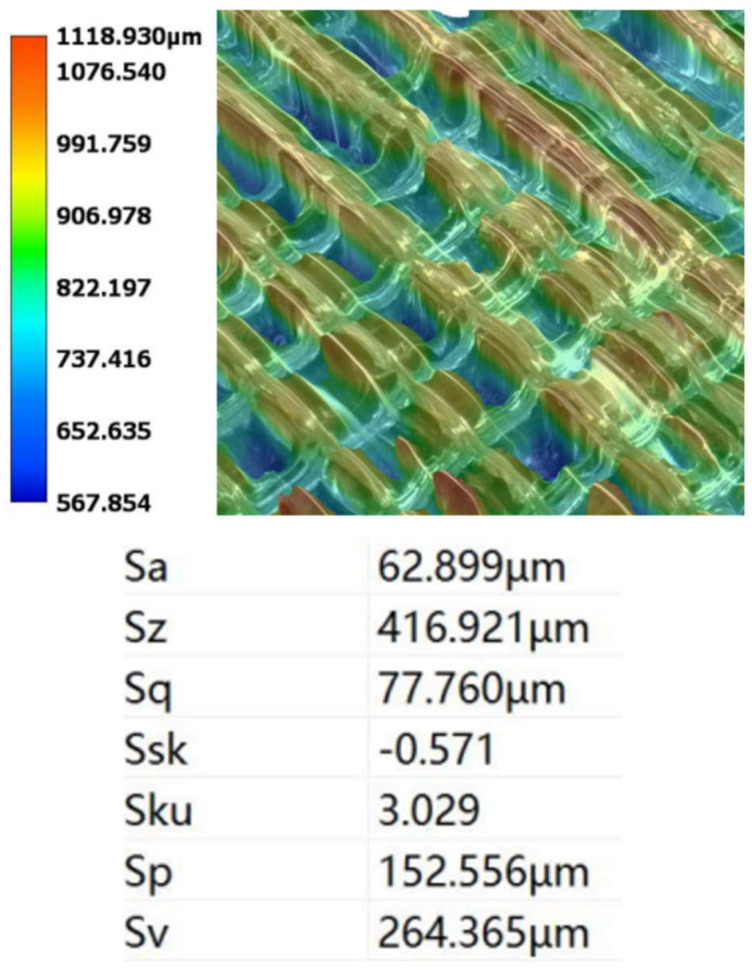
Initial surface roughness of the channel corner region used in the experiment.

**Figure 6 micromachines-16-00987-f006:**
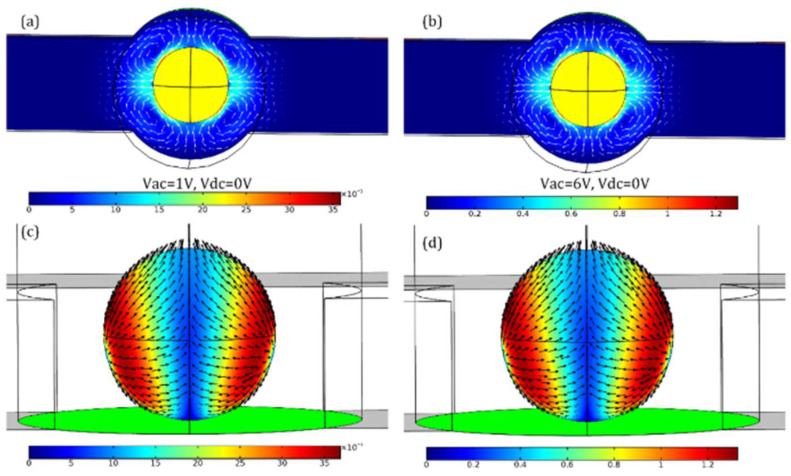
Marangoni chaotic advection induced by bipolar electrocapillary shear stress under AC excitation at 100 Hz. (**a**,**b**) Surface and vector plots of quadrupolar vortex flow fields around a liquid metal droplet (yellow) under different AC voltages: (**a**) $V_{AC}=1\;V$, (**b**) $V_{AC}=6\;V$. (**c**,**d**) Symmetric electroconvective streaming observed at the ideally polarizable surface of the droplet under: (**c**) $V_{AC}=1\;V,\left(d\right)\;V_{AC}=6\;V$.

**Figure 7 micromachines-16-00987-f007:**
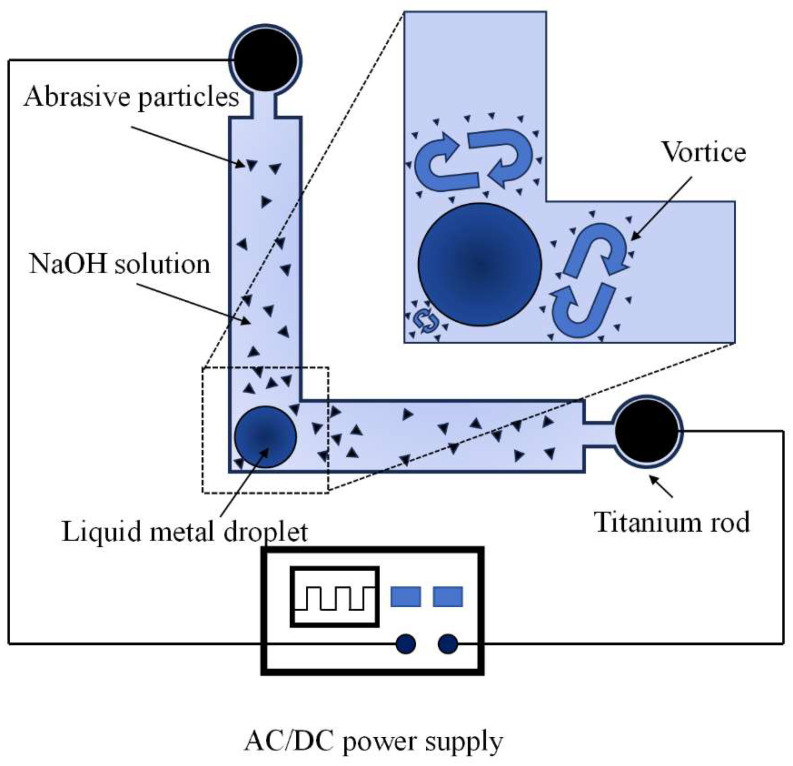
Schematic illustration of localized material removal by liquid metal-driven abrasive flow under an alternating electric field.

**Figure 8 micromachines-16-00987-f008:**
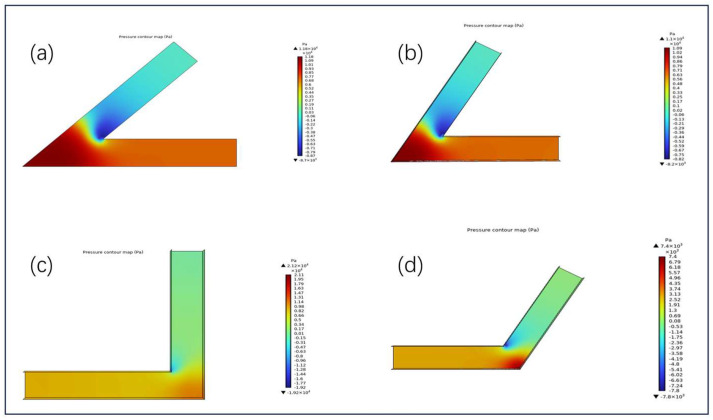
Pressure distribution in microchannels with different bend angles: (**a**) 45°, (**b**) 60°, (**c**) 90°, and (**d**) 120°.

**Figure 9 micromachines-16-00987-f009:**
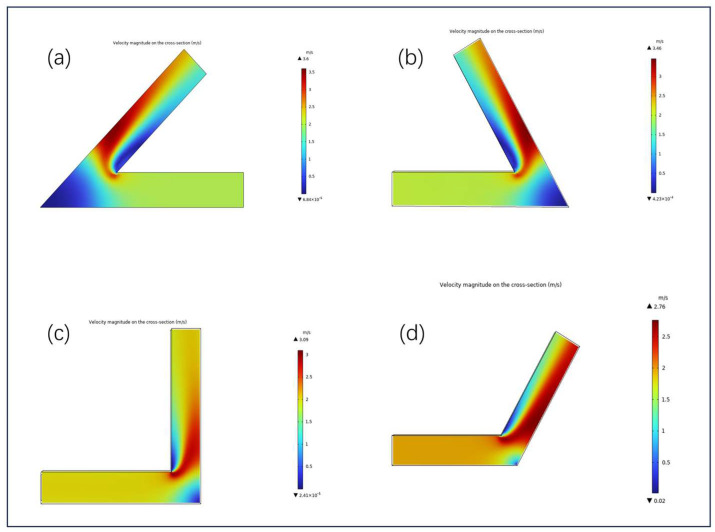
Velocity magnitude distribution under different bend angles: (**a**) 45°, (**b**) 60°, (**c**) 90°, and (**d**) 120°.

**Figure 10 micromachines-16-00987-f010:**
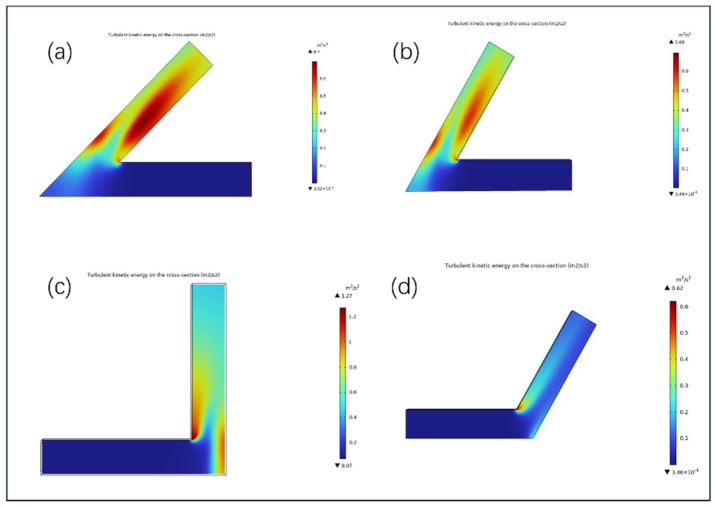
Turbulent kinetic energy distribution in channels with different bend angles: (**a**) 45°, (**b**) 60°, (**c**) 90°, and (**d**) 120°.

**Figure 11 micromachines-16-00987-f011:**
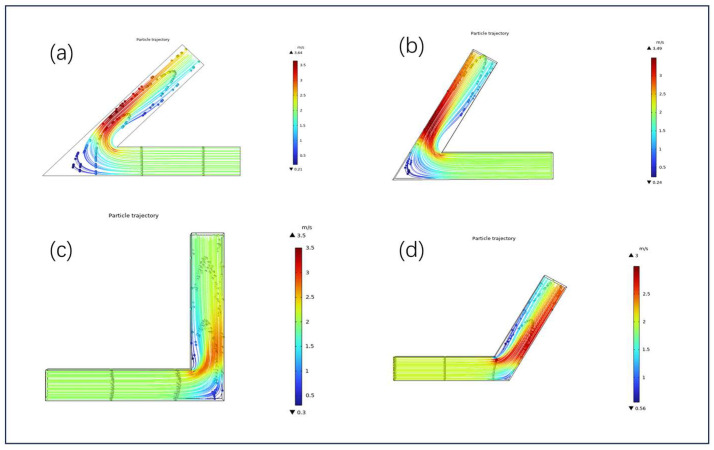
Particle trajectories under different corner angles: (**a**) 45°, (**b**) 60°, (**c**) 90°, and (**d**) 120°.

**Figure 12 micromachines-16-00987-f012:**
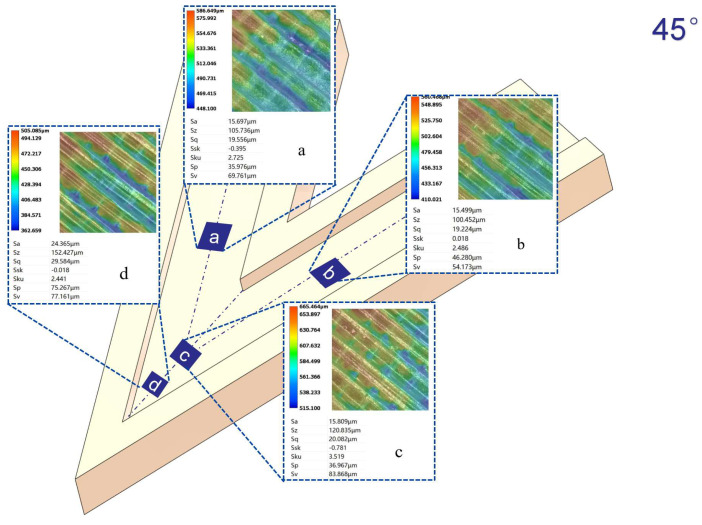
Surface roughness Sa (µm) at different regions of the 45° channel after abrasive flow, (**a**) Sa value in region a; (**b**) Sa value in region b; (**c**) Sa value in region c; (**d**) Sa value in region d.

**Figure 13 micromachines-16-00987-f013:**
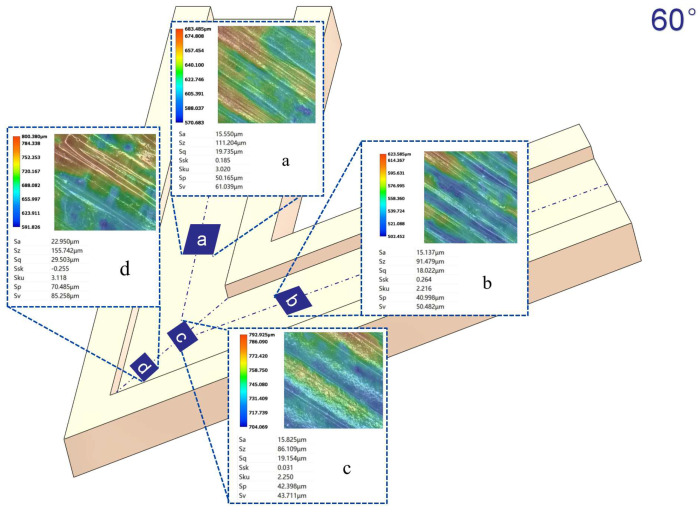
Surface roughness Sa (µm) at different regions of the 60° channel after abrasive flow machining, (**a**) Sa value in region a; (**b**) Sa value in region b; (**c**) Sa value in region c; (**d**) Sa value in region d.

**Figure 14 micromachines-16-00987-f014:**
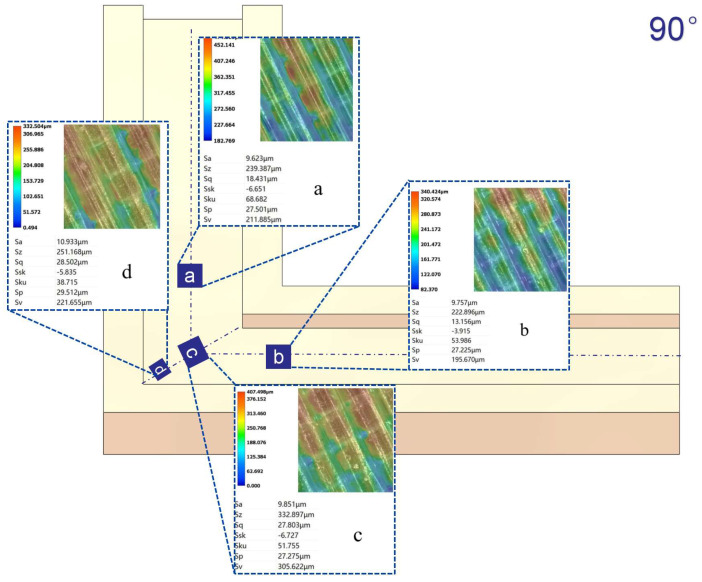
Surface roughness Sa (µm) at different regions of the 90° channel after abrasive flow machining, (**a**) Sa value in region a; (**b**) Sa value in region b; (**c**) Sa value in region c; (**d**) Sa value in region d.

**Figure 15 micromachines-16-00987-f015:**
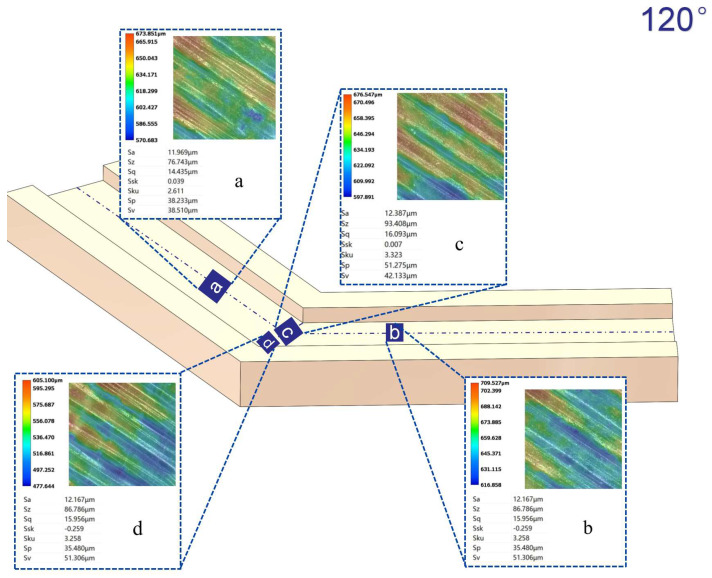
Surface roughness Sa (µm) at different regions of the 120° channel after abrasive flow machining, (**a**) Sa value in region a; (**b**) Sa value in region b; (**c**) Sa value in region c; (**d**) Sa value in region d.

**Figure 16 micromachines-16-00987-f016:**
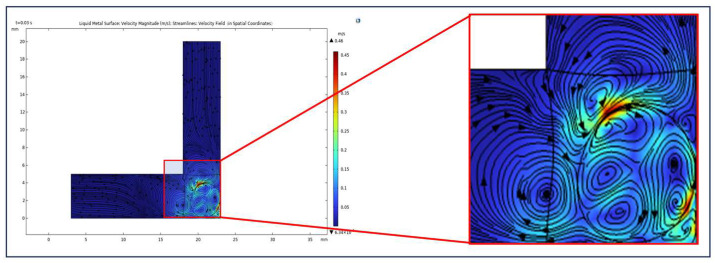
Local vortex field induced by bipolar electrocapillary shear around the liquid metal droplet under AC field.

**Figure 17 micromachines-16-00987-f017:**
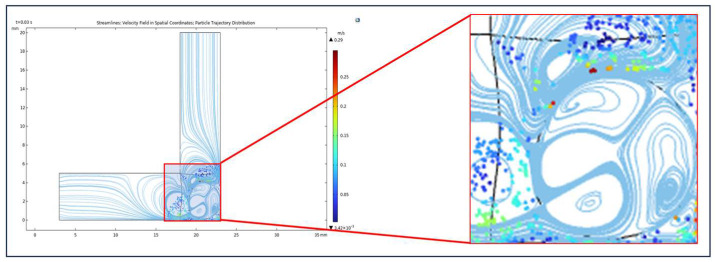
Enhanced particle trajectory redistribution near the corner under quadrupolar vortex flow excitation.

**Figure 18 micromachines-16-00987-f018:**
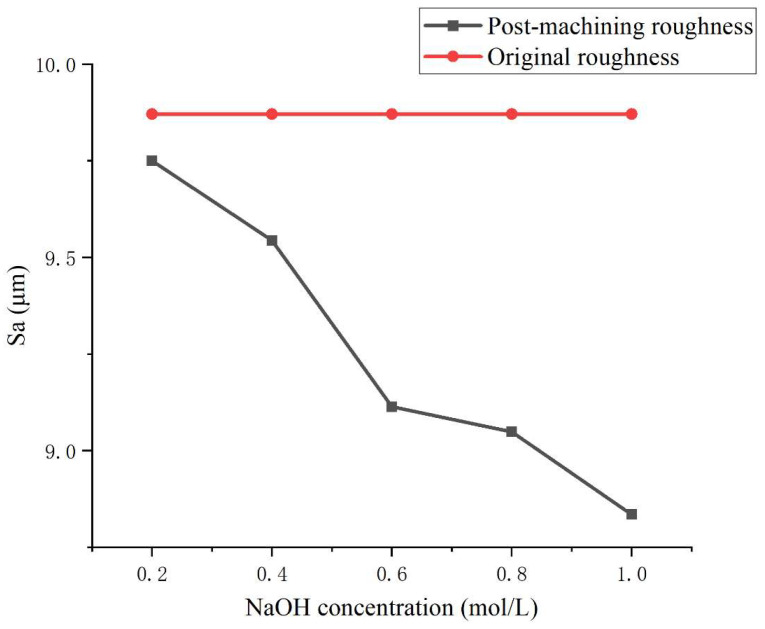
Variation in surface roughness (Sa) with different NaOH concentrations during LM-AF polishing at a 90° channel corner.

**Figure 19 micromachines-16-00987-f019:**
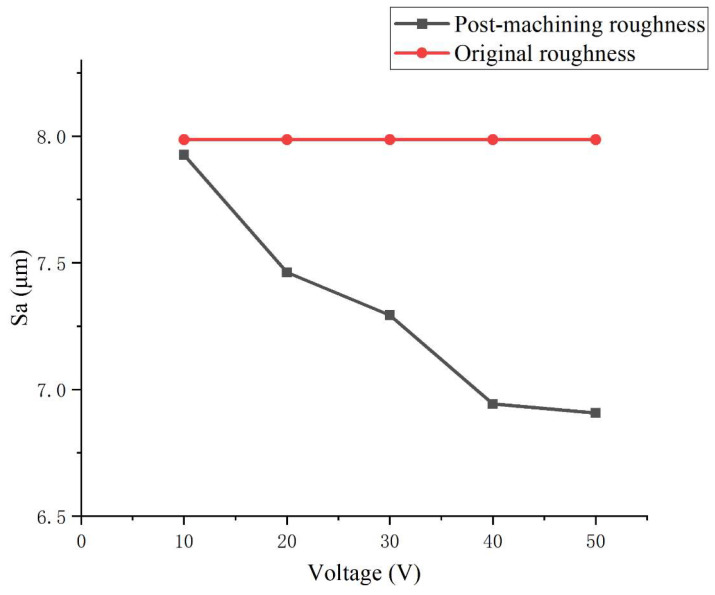
Effect of applied AC voltage on surface roughness (Sa) in LM-AF polishing of a 90° channel corner.

**Figure 20 micromachines-16-00987-f020:**
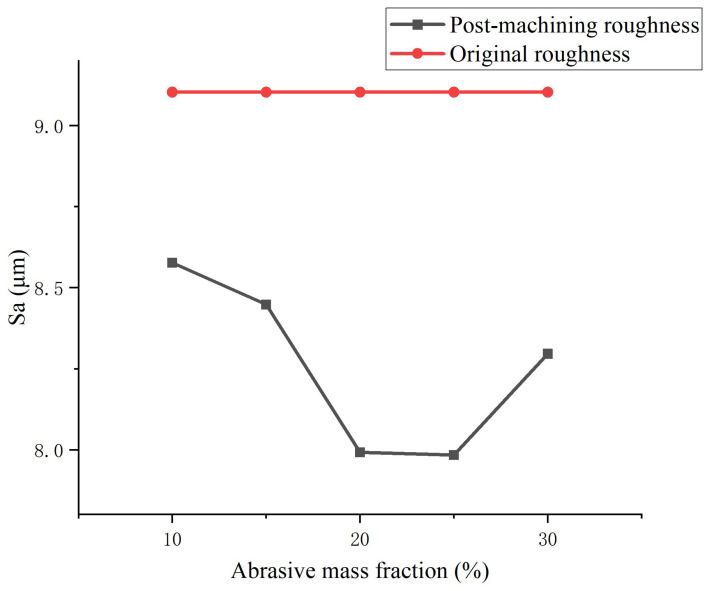
Influence of abrasive mass fraction on the surface roughness (Sa) after LM-AF polishing at a 90° channel corner.

**Figure 21 micromachines-16-00987-f021:**
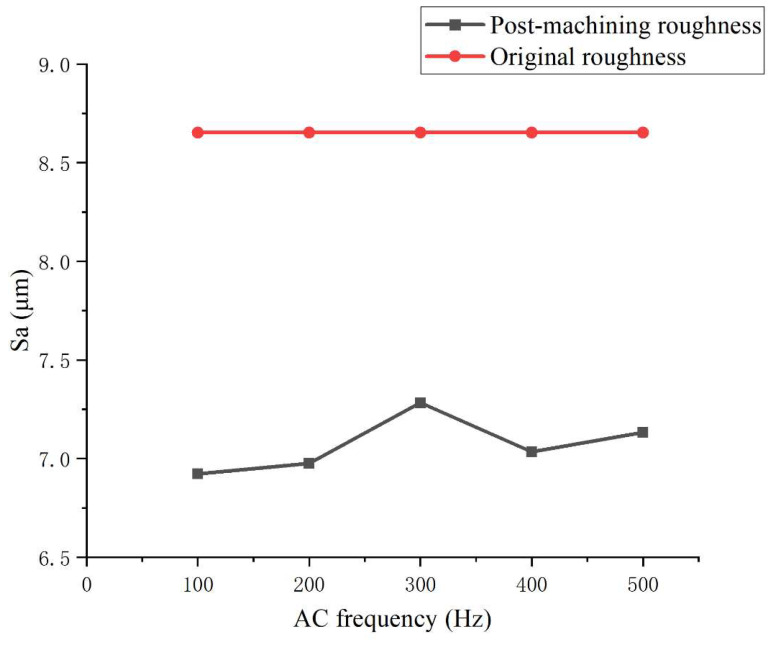
Effect of AC frequency on surface roughness (Sa) during LM-AF polishing at a 90° channel corner.

**Figure 22 micromachines-16-00987-f022:**
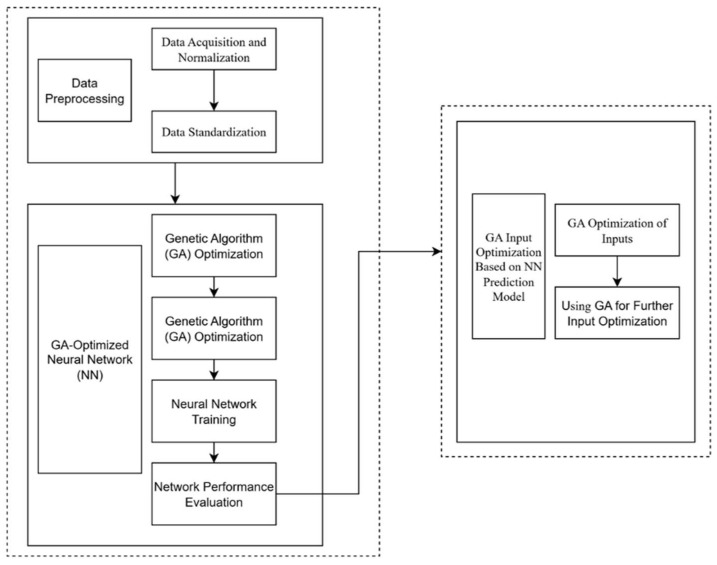
Algorithm flowchart for GA-NN-GA optimization process.

**Figure 23 micromachines-16-00987-f023:**
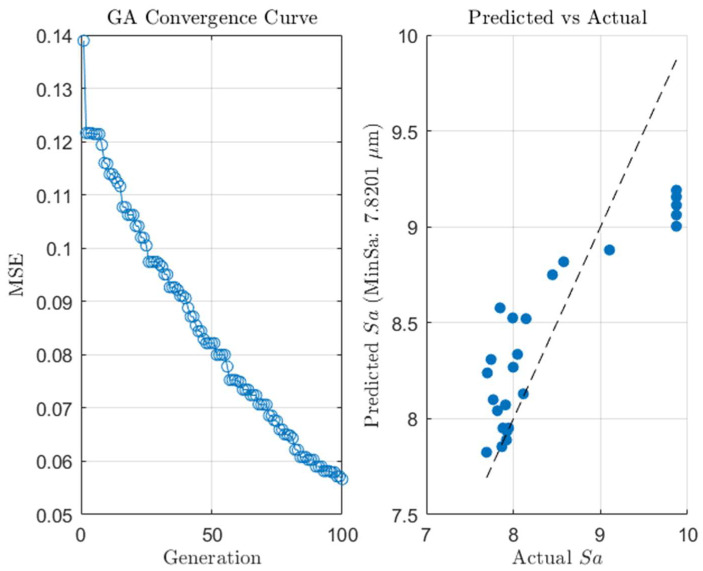
GA optimization process convergence curve and predicted vs. actual results for surface roughness (Sa). The **left** panel shows the convergence of the genetic algorithm (GA) over generations, where the MSE (Mean Squared Error) gradually decreases, indicating the model is approaching optimal parameters. The **right** panel presents a comparison between predicted and actual surface roughness values $\left(Sa\right),$ with a minimum predicted Sa of 7.8201 μm, demonstrating the accuracy of the GA-based optimization model in predicting Sa.

**Figure 24 micromachines-16-00987-f024:**
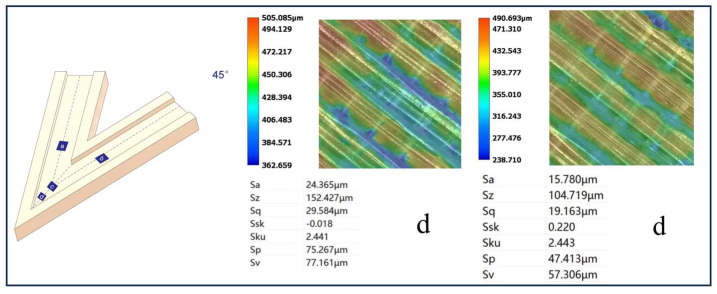
Surface morphology of the 45° channel before and after polishing (left: before polishing).

**Figure 25 micromachines-16-00987-f025:**
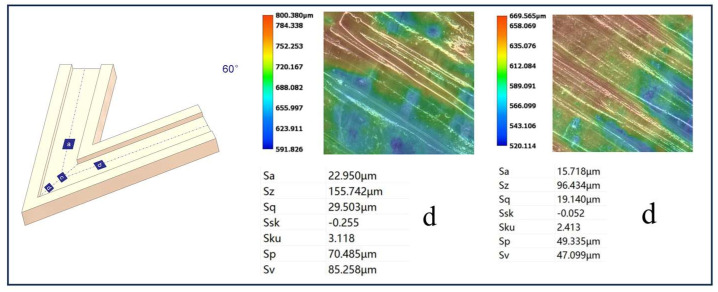
Surface morphology of the 60° channel before and after polishing (left: before polishing).

**Figure 26 micromachines-16-00987-f026:**
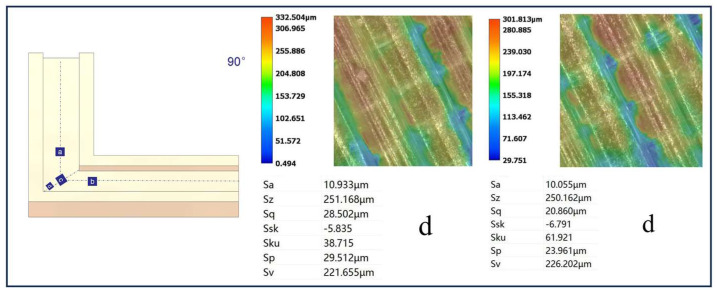
Surface morphology of the 90° channel before and after polishing (left: before polishing).

**Table 1 micromachines-16-00987-t001:** Basic simulation parameters.

Simulation Parameters	Value
Channel Material	PETG
$\mathrm{Fluid}\;\mathrm{Density}\;\rho_{L1}/\left(kg\cdot m^{-3}\right)$	1000
$\mathrm{Abrasive}\;\mathrm{Density}\;\rho_{L2}/\left(kg\cdot m^{-3}\right)$	3200
Average Abrasive Diameter D/μm	61
Abrasive Volume Fraction V(%)	10
Inlet Condition	Velocity controlled, normal flow 2 m/s (fully developed)
Exit Condition	Pressure controlled (*p* = 0), prevent backflow

**Table 2 micromachines-16-00987-t002:** Process parameters.

Process Parameters	Value
Abrasive Grain Type	SiC
Liquid phase	Dimethylsilicone oil
Grain size	240
Grain volume fraction (%)	30
Processing Time (h)	16
Viscosity of abrasive slurry (mPa⋅s)	308

**Table 3 micromachines-16-00987-t003:** Simulation parameters.

Simulation Parameters	Value
Abrasive grain type	B_4_C
Grain size	625
Radius of liquid metal (cm)	4
Frequency (Hz)	100
Applied Voltage V0/V	40
$\mathrm{Liquid}\;\mathrm{Metal}\;\mathrm{droplet}\;\mathrm{Density}\;\rho_{L3}/\left(kg\cdot m^{-3}\right)$	6400
NaOH concentration (mol/L)	0.8

**Table 4 micromachines-16-00987-t004:** Design scheme of single-factor polishing experiments.

Group No.	NaOH Concentration (A)	Applied Voltage Amplitude (B)	AC Field Frequency (C)	Abrasive Concentration (D)
1	0.2/0.4/0.6/0.8/1	40	100	20
2	0.8	10/20/30/40/50	100	20
3	0.8	40	100/200/300/400/500	20
4	0.8	40	100	10/15/20/25/30
5	0.8	40	100	20

**Table 5 micromachines-16-00987-t005:** Orthogonal experimental factors and levels.

	Factors	A	B	C	D
Levels		NaOH Concentration (mol/L)	Applied Voltage Amplitude (V)	AC Field Frequency (Hz)	Abrasive Concentration (%)
1		0.2	10	100	10
2		0.4	20	200	15
3		0.6	30	300	20
4		0.8	40	400	25
5		1	50	500	30

**Table 6 micromachines-16-00987-t006:** L25(54) Orthogonal experimental arrangement table.

Group No.	NaOH Concentration (A)	Applied Voltage Amplitude (B)	AC Field Frequency (C)	Abrasive Concentration (D)
1	0.2	10	100	10
2	0.2	20	200	15
3	0.2	30	300	20
4	0.2	40	400	25
5	0.2	50	500	30
6	0.4	10	200	20
7	0.4	20	300	25
8	0.4	30	400	30
9	0.4	40	500	10
10	0.4	50	200	15
11	0.6	10	300	30
12	0.6	20	400	10
13	0.6	30	500	15
14	0.6	40	200	20
15	0.6	50	300	25
16	0.8	10	400	15
17	0.8	20	500	20
18	0.8	30	200	25
19	0.8	40	300	30
20	0.8	50	400	10
21	1	10	500	25
22	1	20	100	30
23	1	30	200	10
24	1	40	300	15
25	1	50	400	20

**Table 7 micromachines-16-00987-t007:** Experimental results.

Group No.	NaOH Concentration (A)	Applied Voltage Amplitude (B)	AC Field Frequency (C)	Abrasive Concentration (D)	Sa (μm)
1	0.2	10	100	10	9.871
2	0.2	20	200	15	9.871
3	0.2	30	300	20	9.871
4	0.2	40	400	25	9.871
5	0.2	50	500	30	9.871
6	0.4	10	200	20	9.103
7	0.4	20	300	25	8.577
8	0.4	30	400	30	8.448
9	0.4	40	500	10	7.992
10	0.4	50	200	15	7.847
11	0.6	10	300	30	8.145
12	0.6	20	400	10	8.047
13	0.6	30	500	15	7.997
14	0.6	40	200	20	7.743
15	0.6	50	300	25	7.7
16	0.8	10	400	15	8.114
17	0.8	20	500	20	7.909
18	0.8	30	200	25	7.767
19	0.8	40	300	30	7.816
20	0.8	50	400	10	7.944
21	1	10	500	25	7.925
22	1	20	100	30	7.879
23	1	30	200	10	7.92
24	1	40	300	15	7.867
25	1	50	400	20	7.692

**Table 8 micromachines-16-00987-t008:** Evaluation metric results for Sa predicted by GA-NN-GA.

Evaluation Indicators	Surface Roughness (Sa)
$R^{2}$	0.99642
RMSE	0.46282
MAE	0.335895

**Table 9 micromachines-16-00987-t009:** Surface roughness comparison between conventional AFM and new method.

Sample No.	Channel Angle	Region	Conventional AFM Roughness (Sa,μm)	New Method Roughness (Sa, μm)
1	$45^{\circ }$	Region a	15.697	15.697
2	$45^{\circ }$	Region b	15.499	15.499
3	$45^{\circ }$	Region c	15.809	15.809
4	$45^{\circ }$	Region d	24.365	15.780
5	$60^{\circ }$	Region a	15.550	15.550
6	$60^{\circ }$	Region b	15.137	15.137
7	$60^{\circ }$	Region c	15.825	15.825
8	$60^{\circ }$	Region d	22.950	15.718
9	$90^{\circ }$	Region a	9.623	9.623
10	$90^{\circ }$	Region b	9.757	9.757
11	$90^{\circ }$	Region c	9.851	9.851
12	$90^{\circ }$	Region d	10.933	10.055

## Data Availability

The raw data supporting the conclusions of this article will be made available by the authors upon request.

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
