# Peer review of "Surface Roughness Uniformity Improvement of Additively Manufactured Channels’ Internal Corners by Liquid Metal-Driven Abrasive Flow Polishing"

_micromachines, 2025, doi:10.3390/mi16090987_

Round 1

Reviewer 1 Report

Comments and Suggestions for Authors

The article is very extensive and interesting. Nevertheless, it requires certain revisions in order to improve its clarity and readability. The most important comments are as follows:

  1. The abstract should be improved – please add specific results from the research.

  2. The conclusions should be revised. At present, they contain too much summary and lack concrete findings, including numerical data from the experiments. The conclusions are not specific enough.

  3. Part of the introduction contains a description of the applied method, which should be moved to the following section.

  4. The introduction is too general. Please present more results from previous studies, show key research findings, and clearly indicate the research gaps. At the end of the introduction, please briefly outline the research plan.

  5. A table summarizing the surface roughness values in different regions and in different samples after conventional AFM and the new method is missing. Such a table would allow a direct comparison of these values.

  6. Figures 12–15 are not very clear; the roughness parameter values are hardly visible. I suggest focusing on the Sa parameter, since the other parameters are not analyzed.

  7. In most of the paper, the roughness parameter Sa is used, but in Section 3.4 the authors switch to parameter Ra. This inconsistency makes it impossible to compare the results. Please correct this and use one consistent roughness parameter throughout the paper.

  8. In Figures 24, 25, and 26, the region on the left is marked with d in italics, while on the right it is written in normal font. This introduces ambiguity in the notation. I suggest marking the regions as a and b, as is typical in figures of this type.

  9. A detailed description of the surface roughness measurement method is missing (type of profilometer, measurement range, filtering).

  10. The number of measurement repetitions and the method of data processing are not provided. Statistical analysis is missing.

Author Response

Comment 1:

The abstract should be improved – please add specific results from the research.

Response:

We appreciate the reviewer’s valuable suggestion regarding the abstract. In response to your comment, we have improved the abstract by incorporating specific results from the research. These additions can be found in lines 10 to 41 of the manuscript, where the specific experimental outcomes are highlighted in blue text for clarity.

We believe these modifications enhance the clarity and precision of the abstract, better reflecting the key findings of the study.

Once again, we thank the reviewer for the insightful feedback, and we trust that these revisions improve the overall quality of the manuscript.

Comment 2:

The conclusions should be revised. At present, they contain too much summary and lack concrete findings, including numerical data from the experiments. The conclusions are not specific enough.

Response:

We would like to sincerely thank the reviewer for the constructive feedback regarding the conclusion section. In response to this comment, we have revised the conclusion to provide more specific findings, including detailed experimental data. The revised conclusion now emphasizes the key results of the study, making it more concrete and focused. These changes are reflected in lines 772 to 817 of the manuscript, and the modifications have been highlighted in blue for easy reference.

We believe that these revisions address the concerns raised by the reviewer and improve the clarity and specificity of the conclusion. Thank you again for your valuable suggestions, which have greatly contributed to the enhancement of the manuscript.

Comment 3:

Part of the introduction contains a description of the applied method, which should be moved to the following section.

Response:

We sincerely thank the reviewer for the constructive feedback regarding the inclusion of application methods in the introduction. In response to this valuable comment, we have revised the introduction by removing the detailed description of the application method, which is highlighted in blue text (lines 46-153). This adjustment ensures that the introduction remains focused on the background, key findings, and research gap, while the detailed methodology is presented clearly in the appropriate section.

We believe these revisions have strengthened the clarity and focus of the introduction, aligning with the journal's requirements for manuscript structure. Thank you again for your thoughtful suggestions, which have greatly contributed to improving the manuscript.

Comment 4:

The introduction is too general. Please present more results from previous studies, show key research findings, and clearly indicate the research gaps. At the end of the introduction, please briefly outline the research plan.

Response:

We would like to sincerely thank the reviewer for the valuable suggestion regarding the general nature of the introduction. In response to this comment, we have revised the introduction to provide a more detailed presentation of previous studies, clearly highlighting key research findings and addressing the identified research gaps. Additionally, we have included a brief outline of the research plan at the end of the introduction. These revisions are reflected in the manuscript between lines 46-153, and the changes have been highlighted in blue text for ease of review.

We believe these modifications enhance the clarity and specificity of the introduction, aligning with the reviewer’s recommendations and strengthening the manuscript. Thank you again for your insightful feedback, which has greatly contributed to improving the quality of the paper.

Comment 5:

A table summarizing the surface roughness values in different regions and in different samples after conventional AFM and the new method is missing. Such a table would allow a direct comparison of these values.

Response:

We sincerely thank the reviewer for the valuable suggestion regarding the inclusion of a table summarizing the surface roughness values in different regions and samples. In response to this comment, we have added a table 9 on pages 32-33 of the manuscript, and a detailed description of the table has been provided on page 31 (lines 748-760), highlighted in blue text. This addition allows for a direct comparison of the surface roughness values between conventional AFM and the new method. The table summarizes the results of both methods, making the conclusions more concrete and facilitating a clearer comparison of their performance.

We believe that this modification enhances the clarity and specificity of the manuscript and effectively addresses the reviewer's comment. Thank you again for the insightful feedback, which has greatly contributed to improving the quality of the paper.

Comment 6:

Figures 12–15 are not very clear; the roughness parameter values are hardly visible. I suggest focusing on the Sa parameter, since the other parameters are not analyzed.

Response:

We sincerely thank the reviewer for the valuable feedback regarding Figures 12–15. The issue with the unclear images, particularly the difficulty in viewing the roughness parameter values, was caused by automatic image compression during document insertion, which resulted in a reduction in image quality. We have since resolved this issue by improving the image resolution to ensure that all data is now clearly visible.

Furthermore, as all figures in the manuscript were uploaded as separate files for easier review, this may have led to some inconsistencies in clarity. Regarding the suggestion to focus on the Sa parameter, we would like to clarify that other parameters were measured alongside Sa using the same optical profilometer, but they were not analyzed in detail in this study. The inclusion of these parameters in the figures was to retain the original data, and we kindly ask the reviewer to understand this decision.

Once again, we greatly appreciate the reviewer’s constructive feedback, which has significantly contributed to improving the clarity and presentation of our figures.

Comment 7:

In most of the paper, the roughness parameter Sa is used, but in Section 3.4 the authors switch to parameter Ra. This inconsistency makes it impossible to compare the results. Please correct this and use one consistent roughness parameter throughout the paper.

Response:

We would like to thank the reviewer for pointing out the inconsistency regarding the use of the roughness parameters Sa and Ra. As noted, the Sa parameter is consistently used throughout most of the manuscript; however, in Section 3.4(page 23), the Ra parameter was mistakenly introduced during the writing process.

In response to your comment, we have corrected this inconsistency by revising Section 3.4(page 23) to use the Sa parameter throughout, ensuring uniformity across the entire paper. We apologize for the oversight and appreciate your understanding.

Thank you once again for your valuable feedback, which has helped improve the clarity and consistency of the manuscript.

Comment 8:

In Figures 24, 25, and 26, the region on the left is marked with d in italics, while on the right it is written in normal font. This introduces ambiguity in the notation. I suggest marking the regions as a and b, as is typical in figures of this type.

Response:

We would like to thank the reviewer for the valuable feedback. In response to your suggestion regarding the labeling font in Figures 24, 25, and 26(pages 31-32), we have made the requested modification by changing the italicized labels on the left side to normal font. However, considering that these figures represent pre- and post-polishing comparisons of the same regions, we have decided to continue using the same letter labels (e.g., a, b) to denote the different processing conditions of the same area. This approach facilitates a clearer presentation of the surface changes before and after polishing, while avoiding redundant labels and maintaining consistency and logical coherence in the figures.

We sincerely appreciate the reviewer’s detailed review and insightful suggestions. We believe this modification significantly enhances the clarity of the figure presentation.

Comment 9:

A detailed description of the surface roughness measurement method is missing (type of profilometer, measurement range, filtering).

Response:

We sincerely thank the reviewer for pointing out the need for a more detailed description of the surface roughness measurement method. In response to this valuable comment, we have made the necessary revisions. A comprehensive description of the measurement method, including the type of profilometer, measurement range, and filtering method, has been added in Section 2.6(page 11) of the Materials and Methods (lines 303–321). For clarity, these changes are highlighted in blue text.

We believe these additions address the reviewer’s concerns and enhance the transparency and reproducibility of the experimental methodology. Thank you again for your insightful feedback, which has contributed significantly to improving the manuscript.

Comment 10:

The number of measurement repetitions and the method of data processing are not provided. Statistical analysis is missing.

Response:

We sincerely thank the reviewer for the valuable comment regarding the inclusion of measurement repetition, data processing methods, and statistical analysis. In response to this suggestion, we have revised the manuscript to address these points comprehensively.

Measurement Repetition: We have added details about the measurement repetition in the manuscript on page 26, lines 617-621. Specifically, we conducted three repetitions for each experiment, and the average of these repetitions was used for analysis. This addition is highlighted in blue text for the reviewer's reference.

Data Processing: We have clarified the data processing methods in the manuscript. The data normalization process, as part of the optimization algorithm, was implemented in the code. This is also noted in the manuscript to explain the preprocessing steps applied to the data.

Statistical Analysis: Statistical analysis, including key metrics like , MAE, and RMSE, has been incorporated into the manuscript on pages 27-8, lines 643-654. The analysis includes key statistical metrics such as R², MAE, and RMSE. The detailed results of this analysis, including a comparison of the performance metrics, are described from lines 655 to 664 and are presented in Table 8 for easy reference. These additions are also highlighted in blue text.

We believe these revisions comprehensively address the reviewer's concerns and significantly improve the manuscript's clarity and scientific rigor. Thank you again for your valuable feedback, which has greatly contributed to enhancing the manuscript.

Reviewer 2 Report

Comments and Suggestions for Authors

The paper analyzes how the surface quality of additively manufactured parts can be modified. From the point of view of the study, it can be stated that the element of novelty that could be the integration of the experimental part with the mathematical modeling part.

1.  It should be noted, however, that both the title of the paper and the abstract do not allow us to determine with relative precision the additive manufacturing process and the surfaces that are subject to the surface processing process. For this reason, I recommend modifying both the title and the abstract.    

2. Also, from the point of view of the summary part, it should be noted that if the abbreviations are correctly explained in the beginning, the same cannot be said of line 17 at the end. It is also necessary to modify the summary in compliance with the previously mentioned requirements so that those who will read the study, on the one hand, understand both the subject of the work from the point of view of the elements used, but also to increase the visibility of the study carried out.

3. From studying the paper in its introductory part, it can be observed that after an initial part in which the advantages of additive manufacturing are presented in relation to other manufacturing processes, but also the importance of additive manufacturing, the authors move on to the bibliographic analysis of how the process of modifying the internal quality of the holes in the pipes is carried out and conducted by different researchers. Subsequently, based on the observations made, the advantages of the chosen method are presented and how the authors chose the way to activate the arrangement of particles in relation to the surface that will be subjected to the erosion process. From my point of view, either an unfortunate expression was chosen in line 85 (electric field), or the activation that seems to have a thermal role due to the passage of current through the medium used for surface processing takes into account a more complex physical phenomenon, I think it would be advisable to reanalyze this passage. At the same time, considering the additive manufacturing process used, the quality of the inner surface of the holes is substantially influenced by the kinematic parameters in the layered deposition process of the material, having major influences in the areas of modification of the direction of line generation, but also that of the way in which the layers are generated (vertically, horizontally, or inclined). Due to this fact, I consider it worth reformulating the introduction part from lines 100 to 115 in whole or in part. Regarding the part from lines 116 to 124, it must be pointed out that in the summary there is talk of the Sa parameter, in this part of the introduction there is a reference, I say correct, to the Ra parameter of surface quality, so from this point of view, the summary part also needs to be reconsidered. Last but not least, in this part there is no bibliographical reference to the modeling methodology through graphic simulation, creating the feeling that the mathematical model belongs to the authors. I believe that this part of the introduction should also be reanalyzed and completed in accordance with the previous observation.

4. Regarding the Method and Material, it can be observed that it does not offer specialists sufficient information, both dimensional and additive manufacturing, of the parts that were subjected to the surface quality analysis process. There is only information about the angles chosen for generation, but nothing related to the other dimensional elements, nor about the program with which they were generated. Also, the layering part, the parameters, the additive manufacturing and respectively the manufacturer of the material used for manufacturing and the printer used for manufacturing are not analyzed in order to be able to build on the dimensional accuracy of the generated inner surface roughness. No additive manufactured sample with surface analysis after manufacturing is presented, nor is the one related to the surface analysis after the abrasion process. The only reference to the material used, PETG, is in Table 1. Also, the electrical conductivity properties of the medium used are low, being more of a dielectric medium. The issue arises regarding the thermal process generated by the electric current, which should be explained in more detail. Considering what has been presented, in order to be able to continue analyzing the article, I recommend that the required corrections be made both on the generation side and on the processing side, and then the study be re-evaluated further.    

Author Response

Comment 1:

It should be noted, however, that both the title of the paper and the abstract do not allow us to determine with relative precision the additive manufacturing process and the surfaces that are subject to the surface processing process. For this reason, I recommend modifying both the title and the abstract.  

Response:

We sincerely thank the reviewer for the valuable suggestion regarding the title and abstract. We understand the importance of clearly defining the additive manufacturing process and the surfaces involved in the surface treatment process.

In response to this comment, we have made the following revisions in the manuscript: The title has been modified to "Surface roughness uniformity improvement of additively manufactured channels' internal corners by liquid metal-driven abrasive flow polishing," which more clearly reflects the additive manufacturing process used and the surfaces targeted by the surface treatment process. Additionally, the abstract has been revised accordingly to explicitly express the additive manufacturing process and the surfaces involved in the surface treatment process. These changes are highlighted in blue text to ensure that the abstract is more precise and clearly conveys the core content and methodology of the research, helping readers better understand the study’s objectives.

We believe these changes address the reviewer’s concerns and significantly improve the clarity and precision of the manuscript. Thank you once again for your constructive feedback, which has greatly contributed to enhancing the manuscript.

Comment 2:

Also, from the point of view of the summary part, it should be noted that if the abbreviations are correctly explained in the beginning, the same cannot be said of line 17 at the end. It is also necessary to modify the summary in compliance with the previously mentioned requirements so that those who will read the study, on the one hand, understand both the subject of the work from the point of view of the elements used, but also to increase the visibility of the study carried out.

Response:

We sincerely thank the reviewer for the insightful comment regarding the explanation of abbreviations in the abstract. In response, we have made the following revisions to the manuscript: We have provided clear explanations for the abbreviations "Sa" and "GA-NN-GA" at their first occurrence in the abstract, as per the reviewer’s suggestion. Specifically, "Sa" refers to the surface arithmetic average roughness, and "GA-NN-GA" refers to the hybrid optimization framework combining Genetic Algorithms (GA) and Neural Networks (NN). These explanations have been added at the end of lines 18 and 22, respectively, and are highlighted in blue text for the reviewer’s convenience. These changes ensure that each abbreviation is properly explained when first introduced, enhancing the readability and comprehensibility of the abstract, and allowing readers to better understand the core elements of the study while increasing the visibility of the research.

We believe that these revisions have significantly improved the clarity and specificity of the abstract, allowing readers to more easily grasp the core content of the research. Thank you again for your valuable feedback, which has contributed greatly to improving the manuscript.

Comment 3:

From studying the paper in its introductory part, it can be observed that after an initial part in which the advantages of additive manufacturing are presented in relation to other manufacturing processes, but also the importance of additive manufacturing, the authors move on to the bibliographic analysis of how the process of modifying the internal quality of the holes in the pipes is carried out and conducted by different researchers. Subsequently, based on the observations made, the advantages of the chosen method are presented and how the authors chose the way to activate the arrangement of particles in relation to the surface that will be subjected to the erosion process. From my point of view, either an unfortunate expression was chosen in line 85 (electric field), or the activation that seems to have a thermal role due to the passage of current through the medium used for surface processing takes into account a more complex physical phenomenon, I think it would be advisable to reanalyze this passage. At the same time, considering the additive manufacturing process used, the quality of the inner surface of the holes is substantially influenced by the kinematic parameters in the layered deposition process of the material, having major influences in the areas of modification of the direction of line generation, but also that of the way in which the layers are generated (vertically, horizontally, or inclined). Due to this fact, I consider it worth reformulating the introduction part from lines 100 to 115 in whole or in part. Regarding the part from lines 116 to 124, it must be pointed out that in the summary there is talk of the Sa parameter, in this part of the introduction there is a reference, I say correct, to the Ra parameter of surface quality, so from this point of view, the summary part also needs to be reconsidered. Last but not least, in this part there is no bibliographical reference to the modeling methodology through graphic simulation, creating the feeling that the mathematical model belongs to the authors. I believe that this part of the introduction should also be reanalyzed and completed in accordance with the previous observation.

Response:

1.We sincerely appreciate your valuable comment on the expression used in line 85 of the original manuscript. We recognize that the term "electric field" may not fully capture the more complex physical phenomenon occurring during the surface treatment process.

To clarify, this section is intended to describe the application of an alternating voltage across the two ends of the flow channel, with liquid metal droplets placed in the electrolyte. The current passing through the medium induces the flow of liquid metal, which helps in more evenly distributing the abrasive particles in geometrically constrained regions. The primary role of the electric field is to assist in the uniform distribution of the abrasive particles, rather than considering the thermal effects caused by the current passing through the medium.

In response to your suggestion, we have revised this section in the manuscript from lines 120 to 124, highlighted in blue text, to ensure the description more accurately reflects the scope of this study and avoids any misleading reference to thermal effects.

2.We sincerely appreciate the reviewer’s insightful comment regarding the influence of material layer deposition parameters, particularly the direction of the build lines and the layer orientation (vertical, horizontal, or inclined), on the surface quality of additively manufactured parts. While this study specifically addresses the issue of non-uniform material removal at the internal corners of complex channels fabricated by additive manufacturing, we have not explored how the deposition parameters of the additive manufacturing process—such as the direction of the build lines and layer orientation—affect the surface quality within the channels.

This aspect, which involves investigating the relationship between 3D printing process parameters (especially the build orientation) and surface roughness, is indeed an interesting and important avenue for future research. We acknowledge that the optimization of the forming and post-processing processes in a coordinated manner, as suggested by the reviewer, would provide valuable insights. We plan to pursue this topic in future studies, where we will explore the interplay between additive manufacturing process parameters and post-processing to improve surface quality.

We thank the reviewer for raising this important point, and we will aim to address this in subsequent research.

3.We sincerely thank you for your valuable comment regarding the inconsistency between the Sa surface roughness parameter mentioned in the abstract and the Ra parameter discussed in the introduction. In response to this suggestion, we have made the following revisions to the manuscript:

Clarification of Surface Roughness Parameters: We have simplified the relevant content in the introduction and provided a detailed explanation of the surface roughness parameters in Section 3.4 of the manuscript. We have ensured that the Sa parameter mentioned in the abstract now aligns with the Ra surface roughness parameter discussed in other sections of the paper.

We believe these revisions effectively address the reviewer’s concern and improve the clarity and consistency of the manuscript. Thank you once again for your valuable feedback, which has played a significant role in enhancing the quality of the paper.

4.We sincerely thank you for your valuable comment regarding the lack of references to modeling methods based on graphical simulation in the introduction. We acknowledge that this could have given the impression that the mathematical model presented was entirely original to the authors.

In response to your suggestion, we have added relevant references to the introduction, specifically in line 177 and lines 269-275, where we cite prior works on graphical simulation-based modeling methods. These additions ensure that the appropriate literature is acknowledged and properly cited, enhancing the clarity and foundation of the presented modeling approach.

We believe these revisions effectively address your concern and improve the manuscript’s rigor by situating the current work within the broader context of existing research. Thank you once again for your constructive feedback, which has significantly contributed to improving the quality of the manuscript.

Comment 4:

Regarding the Method and Material, it can be observed that it does not offer specialists sufficient information, both dimensional and additive manufacturing, of the parts that were subjected to the surface quality analysis process. There is only information about the angles chosen for generation, but nothing related to the other dimensional elements, nor about the program with which they were generated. Also, the layering part, the parameters, the additive manufacturing and respectively the manufacturer of the material used for manufacturing and the printer used for manufacturing are not analyzed in order to be able to build on the dimensional accuracy of the generated inner surface roughness. No additive manufactured sample with surface analysis after manufacturing is presented, nor is the one related to the surface analysis after the abrasion process. The only reference to the material used, PETG, is in Table 1. Also, the electrical conductivity properties of the medium used are low, being more of a dielectric medium. The issue arises regarding the thermal process generated by the electric current, which should be explained in more detail. Considering what has been presented, in order to be able to continue analyzing the article, I recommend that the required corrections be made both on the generation side and on the processing side, and then the study be re-evaluated further. 

Response:

1.We sincerely thank you for your thoughtful comment regarding the insufficient information provided in the "Methods and Materials" section. We understand the importance of providing comprehensive details on the dimensions of the additively manufactured parts, the surface quality analysis process, as well as the manufacturing parameters that are critical to understanding the accuracy of surface roughness measurements.

In response to your suggestion, we have revised the manuscript accordingly. Specifically, we have added detailed information about the other dimensional elements, the generation process, and the additive manufacturing parameters, including layer stacking, materials, and printer specifications. These additions, highlighted in blue text, are included in lines 214 to 226 of the manuscript. We believe these revisions address your concerns and provide the necessary clarity for professional readers to fully understand the process used to analyze the surface roughness of the manufactured internal surfaces.

We hope these changes effectively resolve the issue and contribute to the overall clarity and rigor of the manuscript. Thank you once again for your constructive feedback, which has been invaluable in improving the quality of the paper.

2.We sincerely appreciate your valuable comment regarding the lack of surface analysis results for the additively manufactured samples after surface treatment. We understand the importance of presenting the surface condition before and after the polishing process for a comprehensive understanding of the results.

In response to your suggestion, we have made the following revisions: Prior to the polishing experiments, the inner surface of the additively manufactured channel exhibited significant initial roughness due to the inherent characteristics of the printing process. To establish a consistent baseline, the workpiece was pre-polished using sandpaper, and the initial surface roughness was measured to be approximately Sa 62 µm, as shown in Figure 5. This information has been added to the revised manuscript between lines 262 to 265, and is highlighted in blue text for the reviewer’s convenience.

We believe these revisions address your concern and provide clearer insights into the surface characteristics of the additively manufactured samples before and after surface treatment. Thank you once again for your insightful feedback, which has greatly contributed to improving the manuscript.

3.We sincerely appreciate your valuable comment regarding the electrical conductivity properties of the medium and the thermal effects generated by the electric current. We agree that this is a very interesting aspect, especially considering the low electrical conductivity of the medium used in this study, which may indeed influence the surface treatment process.

However, since this study focuses on improving the surface roughness of additively manufactured channels through liquid metal-driven abrasive flow polishing, this issue is beyond the scope of the current research. We have not provided a detailed discussion of the thermal effects induced by the current in the manuscript. Nevertheless, we recognize the importance of this phenomenon and plan to further explore the thermal effects of the electric current in future research. This will be part of a broader research framework, and we will include it in the conclusion section (lines 810 to 817, highlighted in blue text) as a research outlook.

We hope this clarifies the scope of the current study, and we assure you that we will explore this intriguing aspect in future work. Once again, thank you for your valuable feedback, which will provide important guidance for our future research direction.

Round 2

Reviewer 1 Report

Comments and Suggestions for Authors

Thank you for the responses and the revisions made to the manuscript. However, I suggest slightly shortening the abstract. Specific results have been added to the abstract, but it is now too long. I suggest removing text in lines 33 to 41, as this text is very general.

Author Response

Comment 1:

Thank you for the responses and the revisions made to the manuscript. However, I suggest slightly shortening the abstract. Specific results have been added to the abstract, but it is now too long. I suggest removing text in lines 33 to 41, as this text is very general.

Response:

We sincerely appreciate your valuable suggestion regarding the abstract. In response to your feedback, we have made revisions by removing the content from lines 33 to 41, as it was considered too general. Additionally, we have simplified the abstract further (from lines 10 to 27). The revised abstract is now more concise, and all changes have been highlighted in blue text for your convenience.

We believe these revisions have enhanced the clarity and focus of the abstract, making it more succinct while retaining the key information of the study. Once again, we thank you for your insightful feedback, which has played an important role in improving the quality of our manuscript.

Reviewer 2 Report

Comments and Suggestions for Authors

It can be said that the authors have largely implemented the proposed changes. Based on the changes made, it can be said that the study with some additions can be submitted to the final evaluation process.

It would still remain, as I showed in the Materials and Methods section (recommendation 4), to give the actual details of the additive manufacturing process, as I mentioned in the recommendation made at this point.

Personally, from my point of view, it can be considered that the authors of the paper did not actually make the components subject to this study or do not want some researchers to redo the study based on additive manufacturing parameters, or to be able to consider it as the basis for other studies in the field of finishing internal holes in additively manufactured components.

Author Response

Comment 1:

It can be said that the authors have largely implemented the proposed changes. Based on the changes made, it can be said that the study with some additions can be submitted to the final evaluation process.

It would still remain, as I showed in the Materials and Methods section (recommendation 4), to give the actual details of the additive manufacturing process, as I mentioned in the recommendation made at this point.

Personally, from my point of view, it can be considered that the authors of the paper did not actually make the components subject to this study or do not want some researchers to redo the study based on additive manufacturing parameters, or to be able to consider it as the basis for other studies in the field of finishing internal holes in additively manufactured components.

Response:

We sincerely appreciate your valuable feedback. After carefully addressing your comments, we have made revisions to the manuscript. Specifically, we have addressed your concern regarding the need for detailed information on the actual additive manufacturing process. As per your suggestion, we have now included comprehensive details on the manufacturing process, which enhances the understanding of the specific process involved in the dimension and surface quality analysis of additively manufactured components. This addition makes the study more suitable as a foundation for future research on the polishing of internal channels in other additively manufactured components. The relevant content has been added in lines 200 to 218 of the manuscript and highlighted in blue text for your convenience.

We believe that these revisions provide the necessary clarity and allow the study to be considered a more robust basis for future investigations. Thank you once again for your insightful feedback, which has contributed to significantly improving the manuscript.
